# Proteomics reveals the effects of sustained weight loss on the human plasma proteome

Philipp E Geyer[1,2,†], Nicolai J Wewer Albrechtsen[2,3,4,†], Stefka Tyanova[1], Niklas Grassl[1], Eva W Iepsen[3,4], Julie Lundgren[3,4], Sten Madsbad[4,5], Jens J Holst[3,4], Signe S Torekov[3,4] & Matthias Mann[1,2,*]

## Abstract

Sustained weight loss is a preferred intervention in a wide range of metabolic conditions, but the effects on an individual's health state remain ill-defined. Here, we investigate the plasma proteomes of a cohort of 43 obese individuals that had undergone 8 weeks of 12% body weight loss followed by a year of weight maintenance. Using mass spectrometry-based plasma proteome profiling, we measured 1,294 plasma proteomes. Longitudinal monitoring of the cohort revealed individual-specific protein levels with wide-ranging effects of losing weight on the plasma proteome reflected in 93 significantly affected proteins. The adipocyte-secreted SERPINF1 and apolipoprotein APOF1 were most significantly regulated with fold changes of −16% and +37%, respectively ($P < 10^{-13}$), and the entire apolipoprotein family showed characteristic differential regulation. Clinical laboratory parameters are reflected in the plasma proteome, and eight plasma proteins correlated better with insulin resistance than the known marker adiponectin. Nearly all study participants benefited from weight loss regarding a ten-protein inflammation panel defined from the proteomics data. We conclude that plasma proteome profiling broadly evaluates and monitors intervention in metabolic diseases.

**Keywords** diabetes; mass spectrometry; metabolic syndrome; obesity; plasma proteome profiling
**Subject Categories** Metabolism; Post-translational Modifications, Proteolysis & Proteomics; Systems Medicine
**Mol Syst Biol. (2016) 12: 901**

## Introduction

Obesity and the metabolic syndrome represent a major public health burden, predisposing to several diseases including type 2 diabetes and cardiovascular syndromes and increasing the overall likelihood of early death (Eckel *et al*, 2005; Grundy, 2015). The chances of developing the metabolic syndrome can be reduced considerably by sustained weight loss in obese individuals, through its positive effects on a broad range of metabolic risk factors (Hansen & Bray, 2008). However, it is not entirely clear how weight loss exerts these beneficial effects and to what extent they may differ between individuals (Look *et al*, 2013). The metabolic state is reflected in the levels of lipid transport proteins in the blood, most prominently the apolipoprotein family that is involved in lipid turnover. Several apolipoproteins, for instance A1 and B, correlate with cholesterol and triglycerides (Dominiczak & Caslake, 2011). Obesity is also associated with increased systemic low-grade inflammation, as indicated by plasma levels of specific markers such as C-reactive protein (Esser *et al*, 2014). These proteins are normally quantified individually by antibody-based assays, providing only a partial picture of changes in the entirety of proteins in this body fluid, the plasma proteome. In the case of weight loss, particular proteins like sex hormone-binding globulin are known to change (Azrad *et al*, 2012), but a global view of the dynamic changes in the plasma proteome is currently lacking.

Human blood plasma and serum are the predominant matrices for clinical analysis as they are easily accessible and clearly reflect an individual's metabolism. Mass spectrometry (MS)-based proteomics should be an optimal technology to investigate changes in the human plasma proteome, because this holistic approach can in principle yield specific and quantitative information on all proteins in an unbiased way. Due to several technological challenges, including the large "dynamic range" (the difference between most abundant and least abundant proteins), the proteomic analysis of plasma has remained a very specialized endeavor, precluding the analysis of large numbers of individual plasma proteomes (Anderson, 2010, 2014). The technology of MS-based proteomics has drastically improved over the last years (Mann *et al*, 2013; Zubarev & Makarov, 2013; Munoz & Heck, 2014; Aebersold & Mann, 2016), and several groups have reinvestigated the plasma proteome recently (Liu *et al*, 2015; Cominetti *et al*, 2016;

1   Department of Proteomics and Signal Transduction, Max Planck Institute of Biochemistry, Martinsried, Germany
2   NNF Center for Protein Research, Faculty of Health Sciences, University of Copenhagen, Copenhagen, Denmark
3   Department of Biomedical Sciences, Faculty of Health and Medical Sciences, University of Copenhagen, Copenhagen, Denmark
4   NNF Center for Basic Metabolic Research, Faculty of Health and Medical Sciences, University of Copenhagen, Copenhagen, Denmark
5   Department of Endocrinology, Hvidovre University Hospital, Hvidovre, Denmark
    *Corresponding author. Tel: +49 89 8578 2557; E-mail: mmann@biochem.mpg.de
    †These authors contributed equally to this work

Malmstrom *et al*, 2016). Our laboratory has developed an automated, rapid, and robust shotgun proteomics workflow that allows the streamlined analysis of hundreds of plasma proteins from a single drop of blood, a technology that we call "plasma proteome profiling" (Geyer *et al*, 2016). These profiles provide quantitative information on the majority of the classical, functional plasma proteins (Surinova *et al*, 2011), and we speculated that the metabolic status of individuals during weight loss and maintenance would be reflected by their plasma proteomes. We selected a longitudinal prospective cohort (Iepsen *et al*, 2015) from which we measured the plasma proteomes of 43 individuals at seven time points over 14 months. This allowed us to analyze the global changes related to lipid metabolism and inflammatory processes resulting from a fundamental life style change in the plasma proteome for the first time.

# Results

## Measurement of 1,294 plasma proteomes in a weight loss study

We recently described a highly sensitive proteomics sample preparation method that can be performed with a minimum number of steps in a single reaction vial (Kulak *et al*, 2014). On this basis, we subsequently developed an automatable workflow for plasma, which allows the robust measurement of this challenging body fluid in < 1 h (Geyer *et al*, 2016). We reasoned that this technology might enable analysis of relatively large studies, involving longitudinal monitoring of a substantial cohort.

To investigate the biological impact of losing weight on the human plasma proteome, we made use of a study in which 52 obese individuals were enrolled for an 8-week-long, diet-induced weight loss intervention of 800 kcal/day during which they lost on average 12% body mass. A total of 43 of these individuals were followed for an additional year of successful weight maintenance (Iepsen *et al*, 2015). We obtained plasma from subjects that were fasted overnight, sampled before and after weight loss as well as at five time points over the subsequent year (Fig 1A).

For the purpose of constructing an MS proteome library, we doubly depleted reference plasma samples from three healthy women and three healthy men for the 20 most abundant plasma proteins. The resulting data files were used to increase the depths of analysis in the subsequent cohort measurements by transferring peptide identifications between liquid chromatography tandem mass spectrometry (LC-MS/MS) runs (Geyer *et al*, 2016). For accurate label-free quantification, we measured quadruplicates of 319 plasma samples of the cohort. The resulting 1,294 plasma proteome measurements (including 18 samples for library construction) represent to our knowledge by far the largest plasma proteomics study in a clinical context (Fig 1B). Data acquisition could be accomplished in a reasonable time (10 weeks), and interestingly, we found that remaining challenging points in the LC-MS/MS measurements were concentrated on the chromatographic rather than the MS side (Fig EV1A).

Across the individuals, we identified 737 plasma proteins (subtracting contaminants such as keratins) and an average of 437 (± 23) per individual. Quantitative accuracy was high as reflected by a mean Pearson correlation coefficient of 0.97 for quadruplicate measurements (Fig EV1B).

As shotgun proteomics is not limited to the analysis of a predefined set of proteins, our measurements contained additional information, for instance, on sample quality (Geyer *et al*, 2016). Erythrocyte lysis, which is indicated by increased levels of highly abundant erythrocyte-specific proteins (HBA1, HBB, HBD, CA1), occurred only in one sample, and minor coagulation events during blood taking were present in five of the 318 samples (Fig EV1C and D). This suggests excellent sample handling procedures throughout the study.

## Plasma protein levels are individual-specific

Study participants were followed longitudinally for 1 year after weight loss, which enabled us to identify individual-specific protein levels as well as intra-individual variability of the plasma proteome profiles. For this analysis, we removed the first two time points, which covered the weight loss intervention (weeks −8 and 0) to minimize weight loss-induced effects. For each of 448 proteins that were quantified in all five time points of at least one individual, we calculated the average level per person and for the entire cohort. Strikingly, 69% of proteins differed from the group average more than twofold and 25% more than fivefold (Fig 2A and Table EV1). To investigate the individual variations across time points, we additionally determined longitudinal protein-specific coefficients of variation (CVs). Volcano plots visualize the proteins with a minimum difference from the group average and a maximum variation in the individual. For instance, in participant 4, pregnancy zone protein (PZP) was 26-fold higher than the group average, but varied less than 10% over time, making it a highly individual-specific protein by these criteria (Fig 2B). For the whole dataset, at a twofold difference and 30% CV cutoff, 46% of all proteins would be individual-specific. This represents a lower limit because any measurement errors would tend to decrease the apparent number of individual-specific proteins. We also note that "individual-specific" contains "sub-group"-specific proteins, because each individual could be a representative of a sub-group.

Overall, protein levels tended to vary considerably between participants, but to remain quite constant over time within each individual, as exemplified by seven individual-specific proteins in Fig 2C. Alpha-2-macroglobulin (A2M) is 10 times higher in some individuals compared to others, but has a mean CV of 6% over the 48 weeks for all of the individuals. The high inter-individual variation in lipoprotein(a) (LPA) over three orders of magnitude has a genetic reason: Individuals vary in the number of LPA kringle domains and secretion into the circulation depends on the size of the protein (Utermann, 1989). Some proteins were detected in only a minority of individuals, for instance the intracellular protein Rab GDP dissociation inhibitor (GDI1/2). It was robustly quantified but only in a single person and may therefore be present in this individual's blood due to tissue leakage. A few proteins including complement factor C3, serum albumin (ALB), vitamin D-binding protein (GC), kininogen-1 (KNG1), hemopexin (HPX), complement factor H (CFH), and clusterin (CLU) show very low variation between individuals and over time (fold difference < 1.3; CV < 20%), indicating tight biological control of their levels (Table EV2).

## A
## Study Design

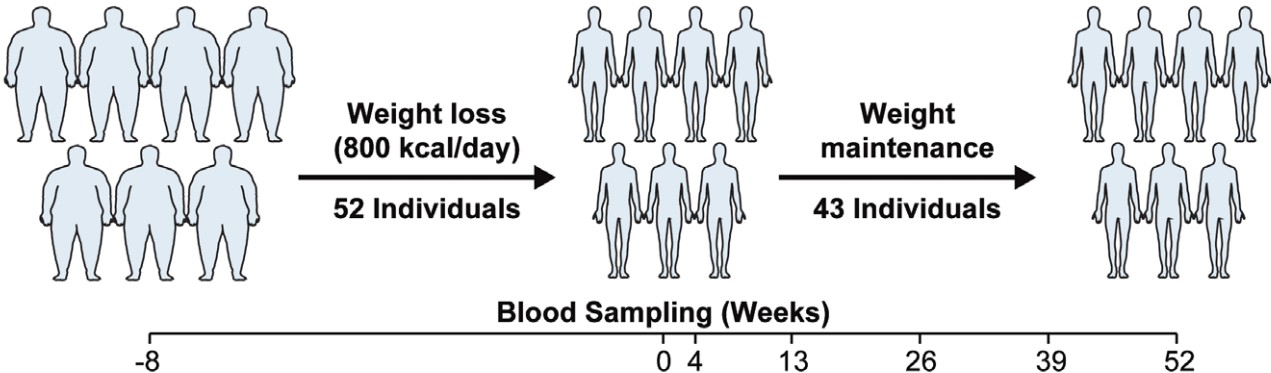

## B
## Proteomic Workflow

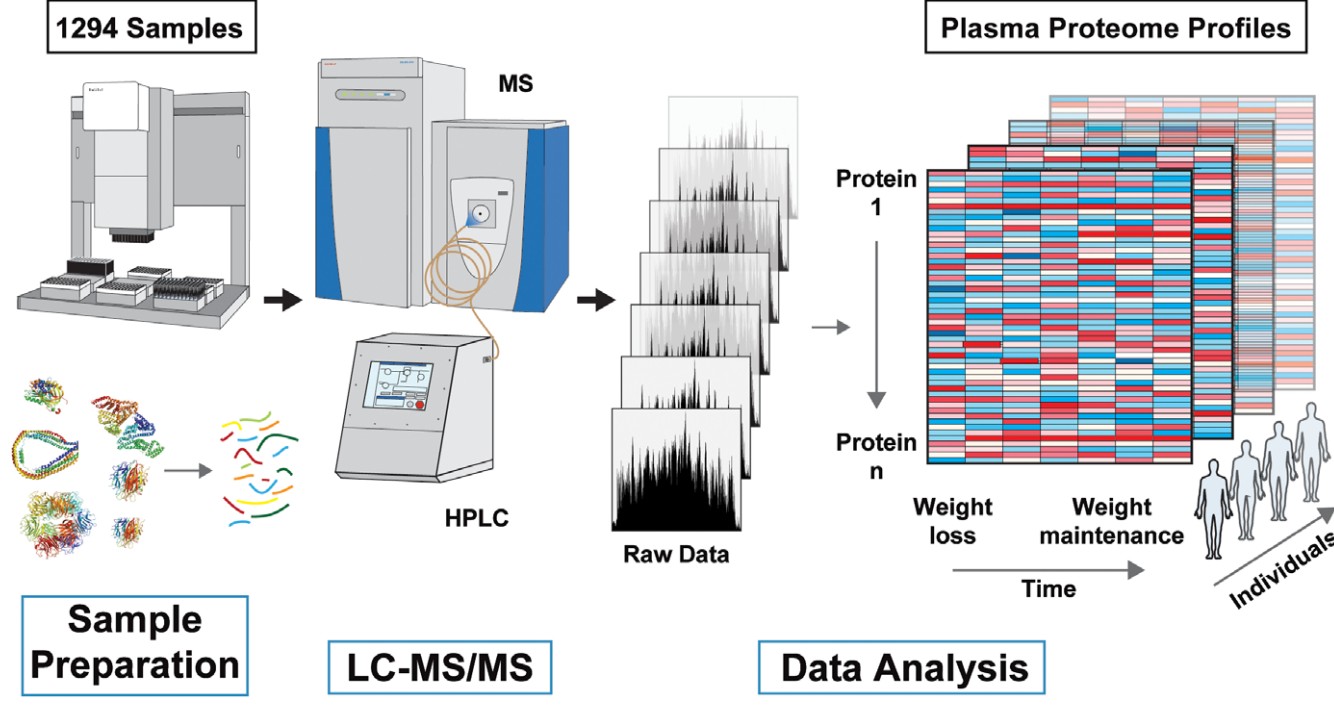

**Figure 1.  Study design and the plasma proteome profile pipeline.**

A   The study cohort consisted of 52 obese individuals, who lost on average 12% of their body mass during 8 weeks of calorie restriction (800 kcal/day). The acute weight loss was followed by a 52-week weight maintenance period by 43 of the study participants with longitudinal blood sampling at the indicated time points.

B   Quadruplicates of the samples and the establishment of a matching library resulted in 1,294 plasma proteomes, which were separately prepared by an automated liquid handling platform. The LC-MS/MS data, which we analyzed by MaxQuant and Perseus, resulted in 319 individual plasma proteome profiles for 52 participants.

### Weight loss changes the plasma proteome profile

Focusing on the effect of weight loss, we analyzed the plasma proteome changes of the study participants from before weight loss to after weight loss (baseline to the 8-week time point). We used a one-sample *t*-test to take individual-specific protein levels into account. Weight loss had a comprehensive systemic effect on the blood plasma proteome profile with 63 decreased and 30 increased protein levels; however, the magnitude of the changes was not large (Fig 3A and Table EV3). For instance, apolipoprotein F (APOF) and inter-alpha-trypsin-inhibitor heavy chain H3 (ITIH3) displayed extremely significantly changing protein levels (both $P < 10^{-13}$) and they increased by 37 and 34%, respectively (Fig 3B). Pigment epithelium-derived factor (SERPINF1) changed with a similar

**A**

| Fold difference | Percentage of all proteins | | | |
|---|---|---|---|---|
| | All | CV<10% | CV<20% | CV<30% |
| >1.2 | 100 | 69 | 92 | 96 |
| >1.5 | 89 | 39 | 71 | 79 |
| >2 | 69 | 15 | 37 | 46 |
| >3 | 45 | 7 | 17 | 21 |
| >4 | 32 | 5 | 10 | 12 |
| >5 | 25 | 3 | 7 | 9 |

**B**

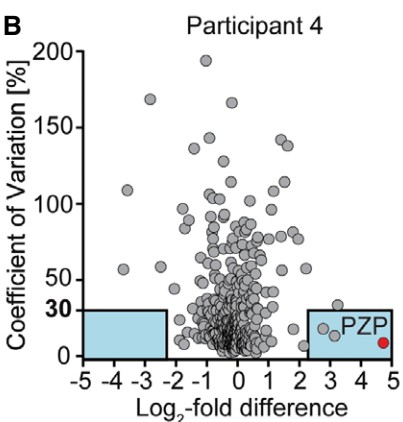

**C**

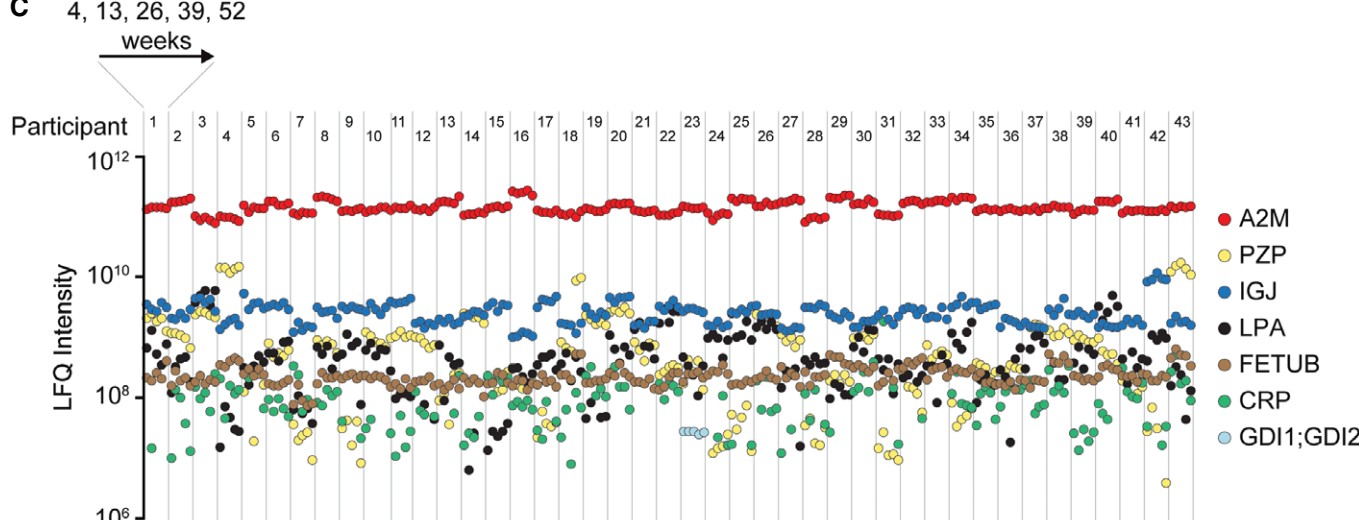

**Figure 2.   Individual-specific plasma protein levels.**

A   Coefficients of variation (CVs) for all proteins were calculated in all participants for the five longitudinal samples. Combination of these longitudinal CVs and fold differences allows estimation of how many proteins are individual-specific.

B   CVs plotted against the $\log_2$-fold difference of one participant compared to the average label-free quantitation (LFQ) intensity of the study cohort. Considering a fivefold difference and a CV below 30% yields the proteins in the blue boxes as specific for this participant.

C   LFQ intensities of seven proteins plotted during weight maintenance (weeks 4–52) for all 43 individuals. These proteins are stable over time within individuals, but strongly vary between individuals. A2M: alpha-2-macroglobulin; PZP: pregnancy zone protein; IGJ: immunoglobulin J chain; LPA: apolipoprotein(a); FETUB: fetuin-B; CRP: C-reactive protein; GDI1/GDI2: Rab GDP dissociation inhibitor alpha/beta.

significance, and here, the average fold difference was only −16%. SERPINF1 is known to be secreted by adipocytes (Wang *et al*, 2004), highlighting the ability of plasma proteomics to pinpoint biologically meaningful but very small changes. Albumin itself, which constitutes about half of plasma proteome mass, increased highly significantly ($P < 10^{-10}$), but only by 8%. Sex hormone-binding globulin (SHBG) changed most strongly due to weight loss with an increase of 117%. A similar effect of weight loss on SHBG levels has been observed before by non-proteomic analysis, serving as a further positive control of our results (Azrad *et al*, 2012). Nevertheless, in all these cases, there are some individuals that deviated from the rest of the cohort. For SHBG, this might be due to its dependence on estrogen levels and thereby age and gender, illustrating the richness of information potentially encoded in the plasma proteome (Geyer *et al*, 2016). Corticosteroid-binding globulin

(SERPINA6), which binds 80% of circulating cortisol, is increased by 12% upon weight loss and together with greater albumin levels may contribute to the decrease in freely circulating cortisol levels upon weight loss (Lewis *et al*, 2005).

## Long-term effect of weight loss due to weight maintenance

Having determined individual-specific and acute weight loss-induced proteins, we next investigated the dynamics of the plasma proteome profile over the 1-year weight maintenance period. We considered proteins that changed highly significantly ($P < 5 \times 10^{-4}$) between baseline and at least one time point after weight loss. Proteins that could have been introduced during the blood sampling procedure, such as keratins, were excluded. In total, 84 proteins fulfilled these criteria. According to their behavior, they clustered into

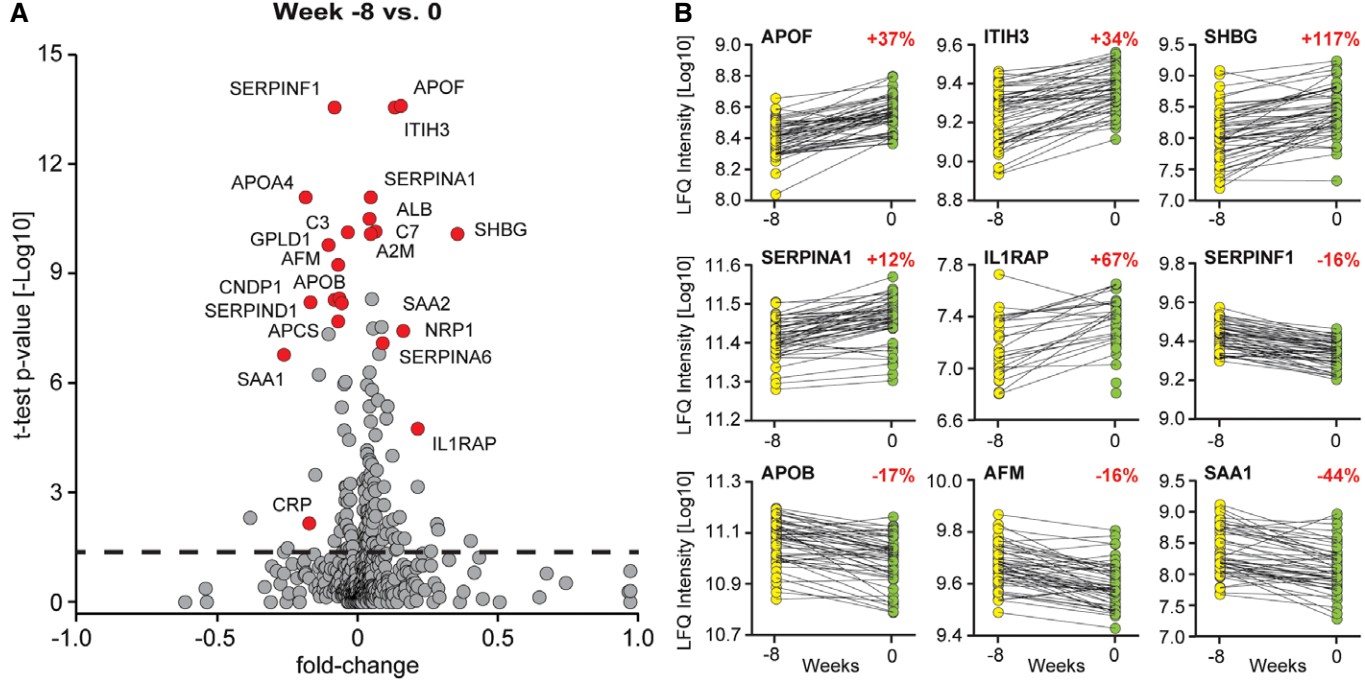

**Figure 3. The effect of weight loss.**

A Volcano plot of proteomes before (week −8) and directly after weight loss (week 0), with the *x*-axis depicting the fold change in protein levels and the *y*-axis the −log₁₀ *t*-test *P*-value of the quadruplicates.

B Changes in protein abundance are shown for each individual. LFQ intensities from before (yellow) and after weight loss (green) are connected by a line to highlight individual-specific protein levels and their changes. The median increase or decrease for each protein over the population is indicated in red.

seven groups, reflecting different aspects of functional adaption to sustained weight loss (Fig 4).

Proteins in the first group decreased rapidly in response to acute weight loss and steadily reverted toward their initial levels over the weight maintenance period. These proteins change transiently in response to the acute energy-deficient state and appear to be regulated toward steady-state levels independent of body weight and fat mass. They included APOA4, afamin (AFM), an alternative transporter for vitamin E, and serum paraoxonase (PON1). PON1, the levels of which are normally determined by enzymatic activity assays, is associated with HDL and is known to exhibit lower activity in obese children and adults (Seres *et al*, 2010; Ceron *et al*, 2014).

Group 2 comprised proteins of special clinical interest as they decreased concomitantly with weight loss and remained low during weight maintenance. They are mainly apolipoproteins (APOA2, APOB, APOC3) and markers of inflammation (CRP, SAA1, SAA4, and ORM2) and will be discussed below. Further members of these two groups were present in a related cluster (group 4) and also exhibited a long-term decrease, but their levels take many months to reach equilibrium. Seven increased and 13 decreased proteins had very significant long-term effects during weight loss (Fig EV2A and B). Apart from the known upregulation of SHBG and downregulation of CRP, novel findings include a near doubling over the entire year of follow-up of the levels of neurophilin-1 (NRP1), whose soluble isoform can bind vascular endothelial growth factor A (VEGFA) (Soker *et al*, 1998). Vitamin D-binding protein (GC) is another protein that was significantly and persistently upregulated.

Proteoglycan 4 (PRG4), normally associated with lubrication of articulating joints, showed a significant long-term decrease upon weight loss (median: −19%), as did heparin cofactor 2 (SERPIND1) (median: −9%), a thrombin inhibitor.

Overall, groups 1, 2, 3 and 7 showed rapid changes in response to lower body weight. In contrast, the adaption of the proteins in groups 4 and 6 was much slower. Zinc-alpha-2-glycoprotein (AZGP1) is an example of a protein that increased slowly over time. It stimulates lipid degradation in adipocytes (Hirai *et al*, 1998), and its sustained elevation may reflect an improved lipid turnover. Likewise, the long-term decrease in the adipocyte-secreted protein SERPINF1 very closely parallels the rapid loss of fat mass followed by long-term maintenance (Fig EV2C).

Related proteins tended to be in the same or related clusters, arguing for the technical reliability of the results and suggesting common and biologically meaningful regulation. For instance, subunits of the same proteins clustered closely together such as C1QA, B, and C. Examples of functionally related proteins include the two serine protease inhibitors SERPINA1 and SERPINA3 (alpha-1-antitrypsin and alpha-1-antichymotrypsin) in group 3 and functionally related members of the complement cascade in group 4 (C4A, C4B, CFI, C8G, C8B, CFB).

The participants of the study had been randomized to treatment with the incretin GLP-1, which showed a significant but small difference in weight loss (Iepsen *et al*, 2015). However, this difference did not correlate with statistically significant changes in our plasma proteome measurements.

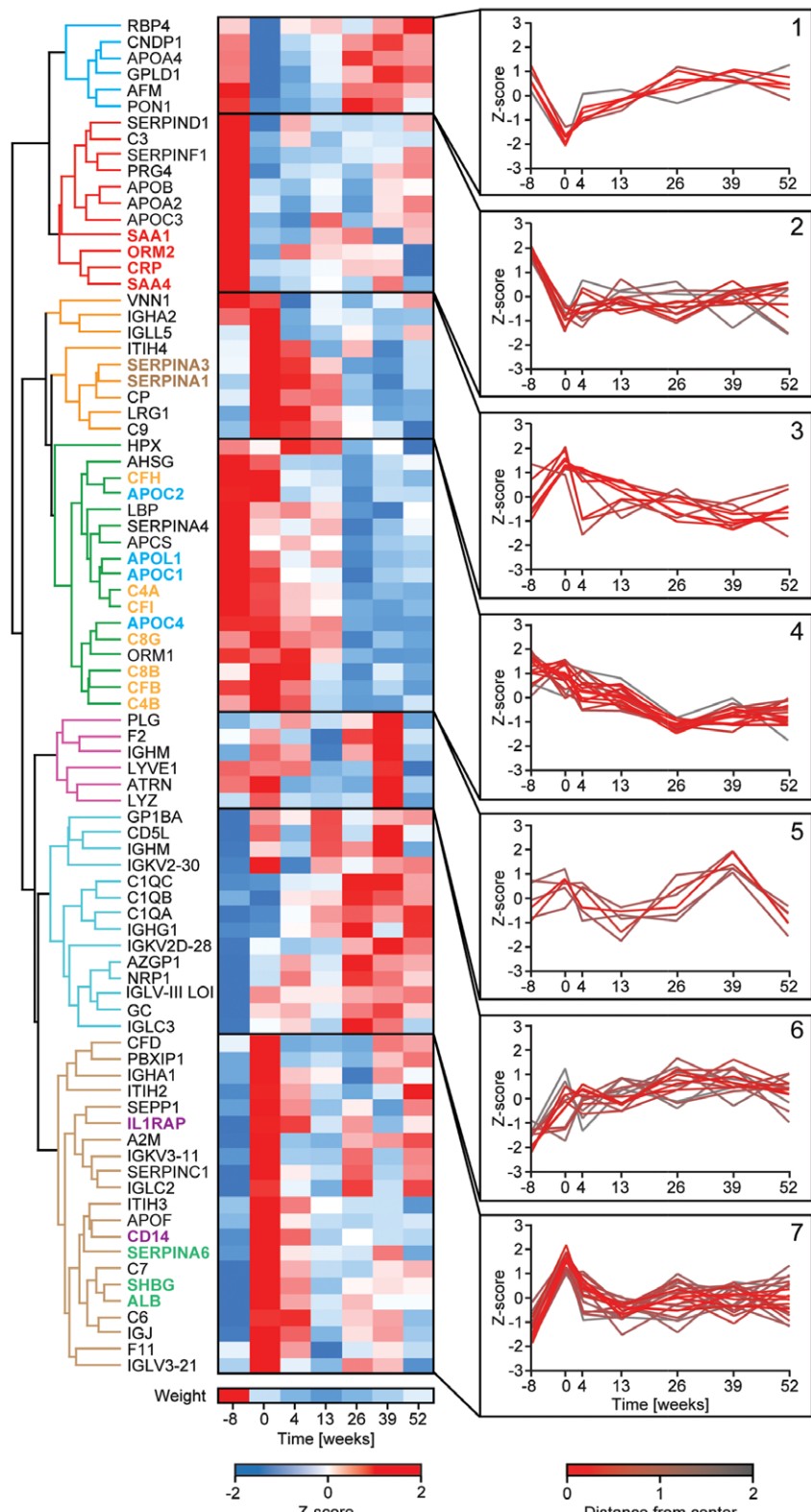

**Figure 4.   Long-term effects of weight loss on the plasma proteome profile.**
Hierarchical clustering of *Z*-scored median LFQ intensities for highly significant proteins (*P* < 0.0005) resulted in seven longitudinal weight regulated protein clusters. Scale bar for *Z*-scored weight is displayed below the protein clusters. Insets show *Z*-scores for proteins in different clusters as a function of time, color-coded for the distance from the center. The highlighted protein names with the same color indicate functionally connected proteins (red: inflammatory markers; brown: serine protease inhibitors; orange: complement system; blue: apolipoproteins; purple: anti-inflammatory-acting proteins; green: steroid transport proteins).

**The effect of weight loss on systemic inflammation factors**

Our results show that weight loss changes multiple components of the plasma proteome. Several acute phase proteins were downregulated. CRP, SAA1, SAA4, and ORM2 cluster closely together in group 2, indicating a fast response upon weight loss (Fig 4). Directly after weight loss, SAA1 and CRP, two prominent risk markers for cardiovascular disease, showed median decreases of 43% and 35%, respectively. ORM1, APCS and LBP decreased more gradually over time (16%, 10% and 16%, respectively, at week 52) and are thus part of group 4. SERPINA1 and SERPINA3 (alpha-1-antitrypsin, alpha-1-antichymotrypsin) are also categorized as acute phase proteins and they are upregulated initially after weight loss (week 0) and start to decline afterward. Altogether, the cluster of highly significant proteins contained a group of 15 complement factors of the classical and alternate complement pathways.

Consistent with decreasing systemic inflammation upon weight loss, the soluble form of the anti-inflammatory protein, interleukin-1 receptor accessory protein (IL1RAP), a known antagonist of the major pro-inflammatory cytokine interleukin-1 (IL-1) (Smith *et al*, 2003), rose upon acute weight loss by an average of 67%. However, this decreased to 18% at the end of the weight maintenance period. Soluble CD14 has been reported to dampen inflammation (Thompson *et al*, 2003), and its levels also increased due to weight loss, but reverted nearly to baseline after 1 year of weight maintenance.

Next, we correlated the quantified plasma proteins with classical laboratory parameters including BMI, HDL, LDL, cholesterol, triglyceride levels, and insulin resistance (HOMA-IR) to investigate whether they were mirrored in the plasma proteome (Fig 5A–F and Table EV4 and see below). Remarkably, of all proteins in our dataset, the five proteins most significantly correlating with BMI were inflammation factors (CFH, C3, APCS, ORM2, and CFI; Fig 5A). For each, the *P*-value was lower than $10^{-8}$ and Pearson correlation coefficients ranged from 0.3 to 0.4. Five other inflammation-related proteins (CRP, SAA4, ORM1, ATRN, and CFB) also correlated significantly with BMI (Table EV5). ATRN (attractin) is a dipeptidase involved in inflammatory responses, but has also been linked to obesity (Duke-Cohan *et al*, 1998; Laudes *et al*, 2010).

Serum amyloid P component (APCS) correlated with HDL, as reported previously (Li *et al*, 1998), but otherwise we found no significant dependency of the above-mentioned inflammatory proteins to other clinical parameters, perhaps because these were only available at three time points (Fig 5B–F).

To calculate a longitudinal systemic inflammation profile for all individuals, we filtered the highly significantly changing proteins (Fig 4) for the keywords "acute phase", "inflammatory response", and "immunity", which resulted in 23 proteins (C1QA/B/C counted as one). From this list, we removed the anti-inflammatory protein CD14 and added two further non-annotated, but known acute phase proteins (APCS and LBP), which were not keyword annotated for the filtered terms. The median MS intensity of the resulting 24 inflammation-related proteins was Z-scored, and we calculated the slope over time for each protein. Of these, 20 inflammation proteins had a negative slope (decreased levels due to weight loss) and 10 further significantly correlated with BMI (Fig 5A and Table EV5). We Z-scored each protein of the ten-protein panel over the individual time series to make them comparable, followed by hierarchical

clustering on the level of the study participants. The resulting heat map is a longitudinal inflammation profile for each of the 42 individuals (Fig 5G).

High levels (red color) are clearly predominant before or directly after weight loss (left side of the heat map), whereas low values are mainly found at the later time points. A group of seven participants is clustered at the top of the heat map and is distinguished by several red patches indicating raised inflammation levels at several time points during weight maintenance. This was not connected to regain of weight, suggesting infection as the cause. For instance, in participant 31, levels of CRP were 28-fold and SAA1 54-fold increased at week 13 compared to her average levels. Focusing on the central 72% of the inflammatory profiles, we calculated the median level of the ten-protein panel and plotted it over time. This revealed that the inflammatory state decreased substantially from before weight loss at week −8 until week 13 and stayed constant at the lower level from then on (Fig 5H).

In addition to these global trends, our proteomic dataset resolves the trajectories of both the individual participants and the individual proteins. For instance, some proteins such as CRP and SAA4 react much faster than others to weight change (Figs 4 and 5G). Moreover, panel values tended to be uniformly high at the beginning and uniformly low at the end, whereas they were more mixed at intermediate time points. To answer the question of how many study participants profited from weight loss regarding their inflammation profile, we averaged Z-scores of the ten-protein panel for each time point and calculated the slope over time. In total, 39 of the 42 individuals had a negative slope, indicating a positive effect on the inflammatory profile of the overwhelming majority of individuals (Table EV6). Investigation of the three individuals that had a positive slope revealed that two gained some weight and the third had high inflammation profile levels at weeks 13 and 26. Moreover, weight regain was present in two further individuals out of five that showed very small positive effects in response to sustained weight loss (Table EV6).

**Proteomic inflammation markers and insulin resistance**

Nearly 40 plasma proteins correlated significantly with HOMA-IR (homeostasis model assessment—insulin resistance). This included adiponectin (ADIPOQ), the protein with the highest known correlation (Weyer *et al*, 2001), but remarkably nine plasma proteins were even more significant (Fig 5F and Table EV4). Excluding an IgG chain, this allowed us to define a positively correlating panel of four proteins (pro-IR) and a negatively correlating panel of five proteins (including adiponectin) (anti-IR).

To compare levels of insulin resistance-related proteins within the study population, we Z-scored each of these proteins along all individuals for each time point. As expected, proteins in the two panels were co-regulated, whereas the panels themselves were anti-regulated (Fig 6A). Nearly all individuals that had high values in the pro-IR panel showed low values in the anti-IR panel. The 24 participants that were in at least one of these groups were considered as individuals with high insulin resistance for the following analyses.

To investigate the known connection between low-grade inflammation and IR (van Greevenbroek *et al*, 2013) at the proteome level, we used the inflammation panel and analyzed it

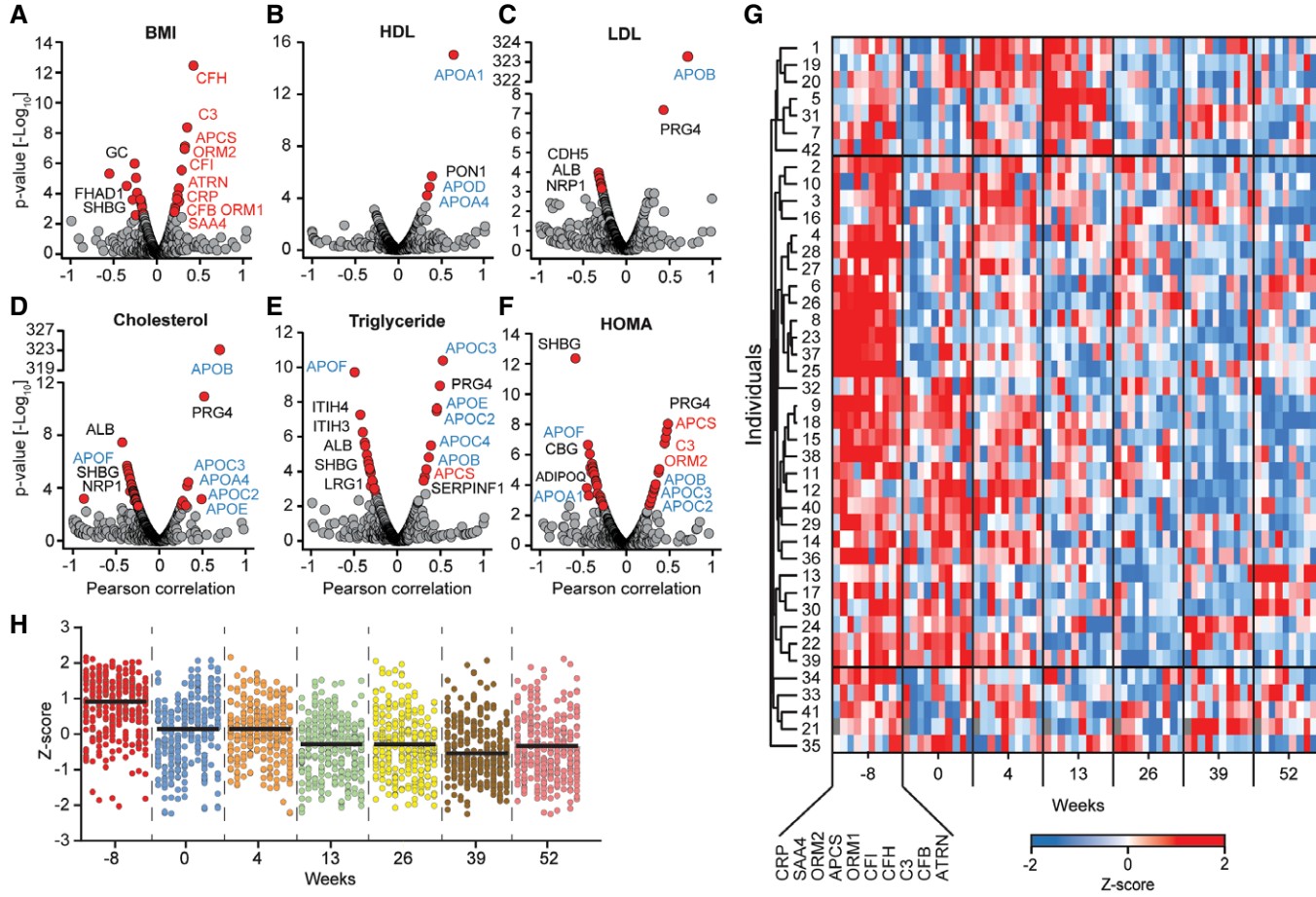

**Figure 5. Plasma proteins and the longitudinal inflammation profile.**

A Correlation of body mass index (BMI) with all quantified proteins in our dataset.

B–F Correlation analysis of the indicated clinical parameter with plasma protein levels over time.

G For each individual, Z-scores were calculated for each of the proteins of the ten-protein panel over the seven time points. The proteins were arranged in the indicated order and hierarchical clustering was performed on the level of the different individuals, resulting in a longitudinal inflammation profile.

H Dot plots for the ten-protein panel in the same order as in (G) for the central cluster. The black line indicates the median Z-score of the inflammation panel for each time point.

Data information: Significant proteins are displayed by red dots, and non-significant ones with gray dots. Red letters indicate inflammation factors that correlate with the BMI and that were used to generate panel (G). A Benjamini–Hochberg FDR of 0.05 was used for significance in all correlation analyses (A–D).

together with the IR panel. We Z-scored the proteins along all individuals and time points, which clearly separated the cohort into higher and lower inflammatory sub-groups. Remarkably, there was an overlap of 14 individuals that were both in the 16-member group with high inflammation and in the 24-member group with the high IR panel values. Thus, a sub-group with high metabolic burden (those with increased plasma levels of markers previously linked to cardiovascular and metabolic diseases) can be determined entirely from the plasma proteome profiling data.

To compare the effects of weight loss and weight maintenance of these 14 individuals to the other 28 study participants, we Z-scored the proteins of the anti-IR/pro-IR and inflammation panel for each protein over the whole study period and all individuals. Both the high- and the comparatively low-risk groups were able to lower their insulin resistance as well as their systemic inflammation levels, as reflected by the panels. The benefit regarding HOMA-IR was even higher for the high metabolic burden group (HOMA-IR:

−19% vs. −39%, Fig 6B). Nevertheless, the high metabolic burden group was only able to adjust the IR and inflammation panels to about the start levels of the low-risk group.

Levels of 80 plasma proteins correlated with leptin levels as determined by ELISA. Inflammation factors like CRP, SAA1, SAA4, C3, CFH, and APCS had highly positive correlations (Table EV4). Six leptin-correlating proteins are part of the ten-protein inflammation panel, which confirms the connection between insulin resistance and inflammation. Leptin positively correlated with all four proteins from the pro-HOMA-IR panel and two of the five proteins of the anti-HOMA-IR panel (NRP1 and APOF) were also anti-correlated with leptin levels.

## Changes in the apolipoprotein family during weight loss

Levels of apolipoprotein family members are of central importance in determining the risk of cardiovascular and metabolic diseases,

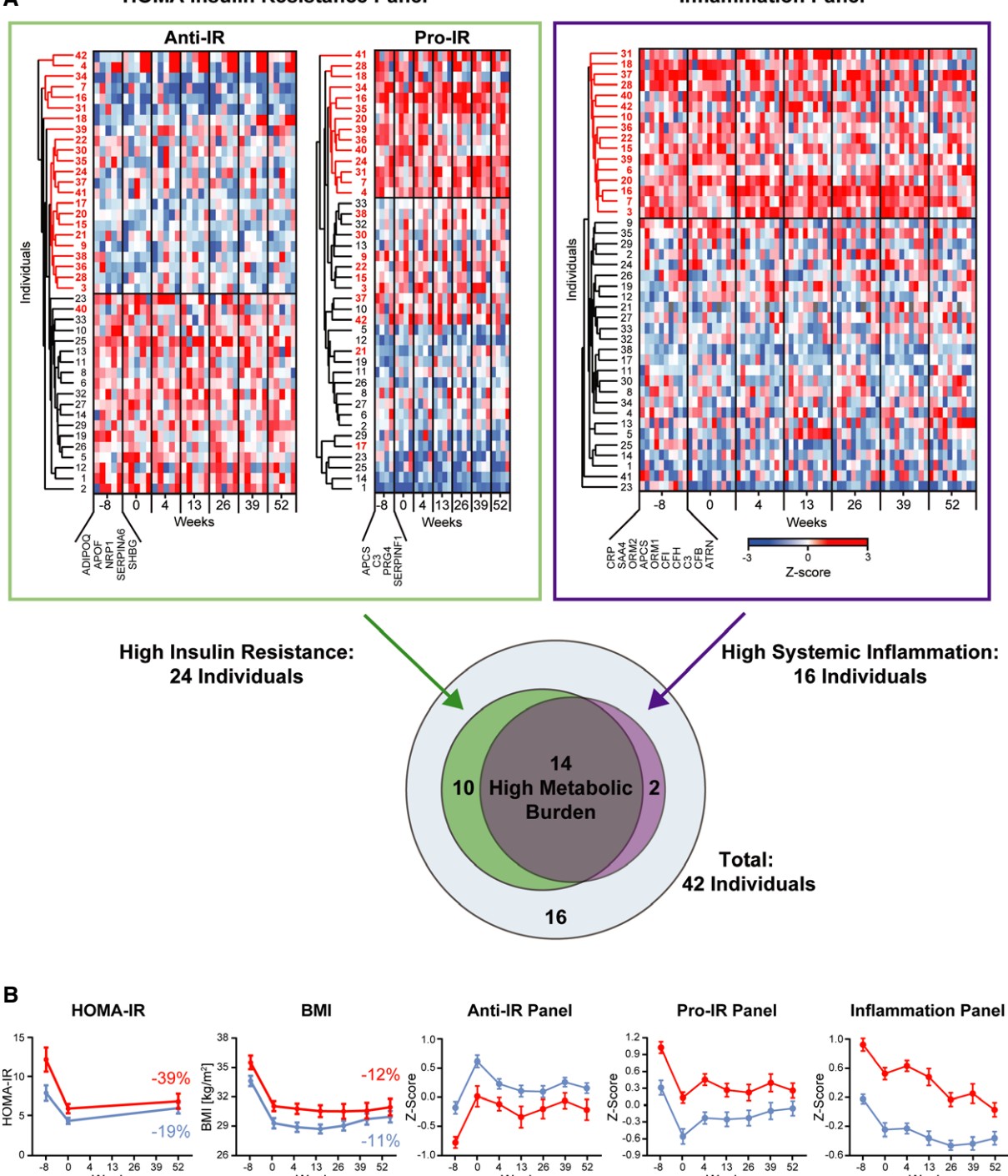

**Figure 6.  Insulin resistance and systemic inflammation.**

A   The five proteins with the highest positive and the four proteins with the highest negative correlation with IR were used to define a pro- and an anti-IR panel. These panels separated the study cohort in a high and a low IR group and the 24 individuals that were present in at least one of the IR panels are highlighted (numbered in red at the *y*-axes). Likewise, the ten-protein inflammation panel separates the cohort in individuals with high and low systemic inflammation levels and individuals with high levels were highlighted (numbered in red). Of 16 study participants with high inflammation levels, 14 were also present in the high IR group as illustrated by a Venn diagram, indicating a high metabolic burden group as defined by plasma proteome profiling.

B   HOMA-IR levels and the BMI are compared between the high and the low metabolic burden group with indicated changes in percent at the study endpoint. These changes are linked to longitudinal changes in the anti-IR, the pro-IR, and the inflammation profile. The means are plotted with SEM as error bars over time.

but current immuno-based assays only measure one or a few of them at a time. In contrast, shotgun proteomics should be able to comprehensively profile the entire family, and indeed, we successfully recorded longitudinal profiles for 18 different apolipoproteins (Fig 7A). Twelve of these changed significantly at least at one point during weight loss or maintenance and six (APOA2, APOB, APOC2, APOL1, SAA1, and SAA4) showed significant long-term effects (Fig EV2A and B). Of the rapid responding apolipoproteins, APOF increased by 37% and APOA4 decreased by 36% upon weight loss, but the levels of both reverted to baseline over the course of a year. In contrast, levels of APOC1, APOC2, and APOC4 consistently decreased and stayed at about 70% of their initial levels. Apolipoprotein(a) (LPA) had the largest absolute change on average as a response to weight loss (increase of 95%).

Next, we correlated the dynamics of the apolipoproteins with BMI, cholesterol, triglyceride, glucose, HDL, and LDL levels. Of these, APOF had a high negative correlation with triglyceride levels ($-0.50$; $P < 6 \times 10^{-8}$) and APOB, APOC2, APOC3, APOC4 as well as APOE a strong positive correlation (0.33, 0.45, 0.52, 0.38, and 0.45, respectively; Fig 5E). APOB, APOC2, APOC3, APOE, and APOF also correlated with total cholesterol (Fig 5D). APOB further strongly correlated with LDL (Fig EV3) (0.72; $P < 3 \times 10^{-323}$), which is expected as each LDL particle contains one APOB molecule (Dominiczak & Caslake, 2011). Similarly, APOA1 is a constituent of HDL and accordingly, it was highly correlated with HDL levels (0.64; $P < 9 \times 10^{-16}$). As mentioned above, APOA4 and PON1 are in the rapid response cluster (group 1 of Fig 4) and both proteins and APOD correlate with HDL measurements (Fig 5B). Several non-apolipoproteins also showed a good correlation with LDL, for instance the above-mentioned PRG4, which furthermore correlated significantly with triglycerides (0.52, 0.48; $P < 3 \times 10^{-11}$, $1 \times 10^{-9}$; Fig 5C).

Interestingly, the ratio of APOB to APOA1, which is used to assess cardiovascular disease risk, decreased due to weight loss by 8% and remained lower over time (week 52: 7%) for 25 of the 42 study participants.

To investigate the general response of lipoprotein particles and metabolic process during weight loss on the basis of the plasma proteome, we used gene ontology (GO). This assigned apolipoproteins to five main lipoprotein particles: chylomicrons, high-density lipoprotein (HDL), intermediate-density lipoprotein (LDL), low-density lipoprotein (LDL), and very low-density lipoprotein particle (vLDL) (Fig 7A and B). Of the 12 apolipoproteins that occur in high-density lipoprotein (HDL) or low-density lipoprotein (LDL) particle, 11 changed significantly. Moreover, we observed a fast response for seven significantly changed apolipoproteins (belonging to clusters 1, 2, and 7 of Fig 4). Globally, the level of the different lipoprotein particles changed most rapidly during weight loss and tended to remain at a lower level during weight maintenance. Performing the same analysis at the level of gene ontology defined "biology processes" likewise showed that most of these that were related to lipoproteins, lipids, cholesterol, and fat decreased with body weight (Figs 7C and EV4). Thus, plasma proteome profiling revealed the dynamics of metabolic changes during weight loss both at the level of individual proteins and at the global levels of lipoprotein particles and processes.

## Discussion

Losing weight and maintaining the weight loss are central topics in modern society, research, and medicine. Although generally viewed as desirable, their effects on cardiovascular and general metabolic risk at the individual level are far from universally agreed (Goodpaster *et al*, 2010; Casazza *et al*, 2013; Look *et al*, 2013; Kushner & Ryan, 2014). Here, we wished to contribute to this debate by deciphering the plasma proteome at a global level, using state-of-the-art MS-based proteomics technologies. We used an automated and robust plasma proteome profiling workflow and successfully measured 1,294 plasma proteomes from 52 obese individuals, revealing dynamic changes in response to 8 weeks of diet-induced weight loss followed by a year of weight maintenance. The depth of proteome coverage obtained here—more than 400 proteins per individual—was sufficient for covering all clinically relevant lipoproteins and markers of low-grade inflammation as well as many other functional blood proteins. Quadruplicate measurements as well as the measurement of time profiles of 43 participants allowed us to pinpoint relatively small changes (< 20%) with very high statistical significance, which compares favorably with standard, antibody-based laboratory tests. Further advantages of MS-based proteomics are that large numbers of proteins can be analyzed simultaneously and with very high specificity, as there is no "cross-reactivity" in MS measurements. Furthermore, measurements are unbiased in the sense that the identity of analytes does not have to be known beforehand. We found that the global nature of plasma proteomics also allows us to quickly assess the quality of individual samples and entire studies on the basis of erythrocyte lysis markers and proteins involved in coagulation (Fig EV1).

Omics technologies have already been brought to bear on the study of obesity. GWASs have linked specific loci with genetic propensity for this trait, whereas transcriptome studies have investigated tissues such as fat, muscle, or white blood cells. These studies are by their nature not directly connected to changes in protein levels in the plasma. In contrast, metabolomics has been performed on plasma in the context of weight loss, which demonstrated that several metabolic markers change after weight loss and similar to what we have reported here that individual-specific levels seem to predominate (Piccolo *et al*, 2015; Wahl *et al*, 2015; Newgard, 2016). At the protein level, individual regulators of lipid transport have been studied in depth, but our study provides a first proteomic view of changes in the plasma. Although not subject of this study, it would be interesting in the future to connect the different omics datasets to obtain a more comprehensive understanding of physiological changes during weight loss and maintenance.

Our study design allowed us to separate the influence of weight loss and weight maintenance on the plasma proteome and also provided a systematic view of the variations in the levels of hundreds of plasma proteins in a human cohort. We defined "individual-specific protein levels" as those whose variation over time in each individual was small compared to the difference between the individuals. By these criteria, a surprisingly large part of the plasma proteome was individual- or sub-group-specific as nearly half of the proteins varied more than twofold, while their longitudinal CV was less than 30%. Thus, levels of many proteins remain essentially similar over long time periods, but vary between different individuals. Such observations have already been made

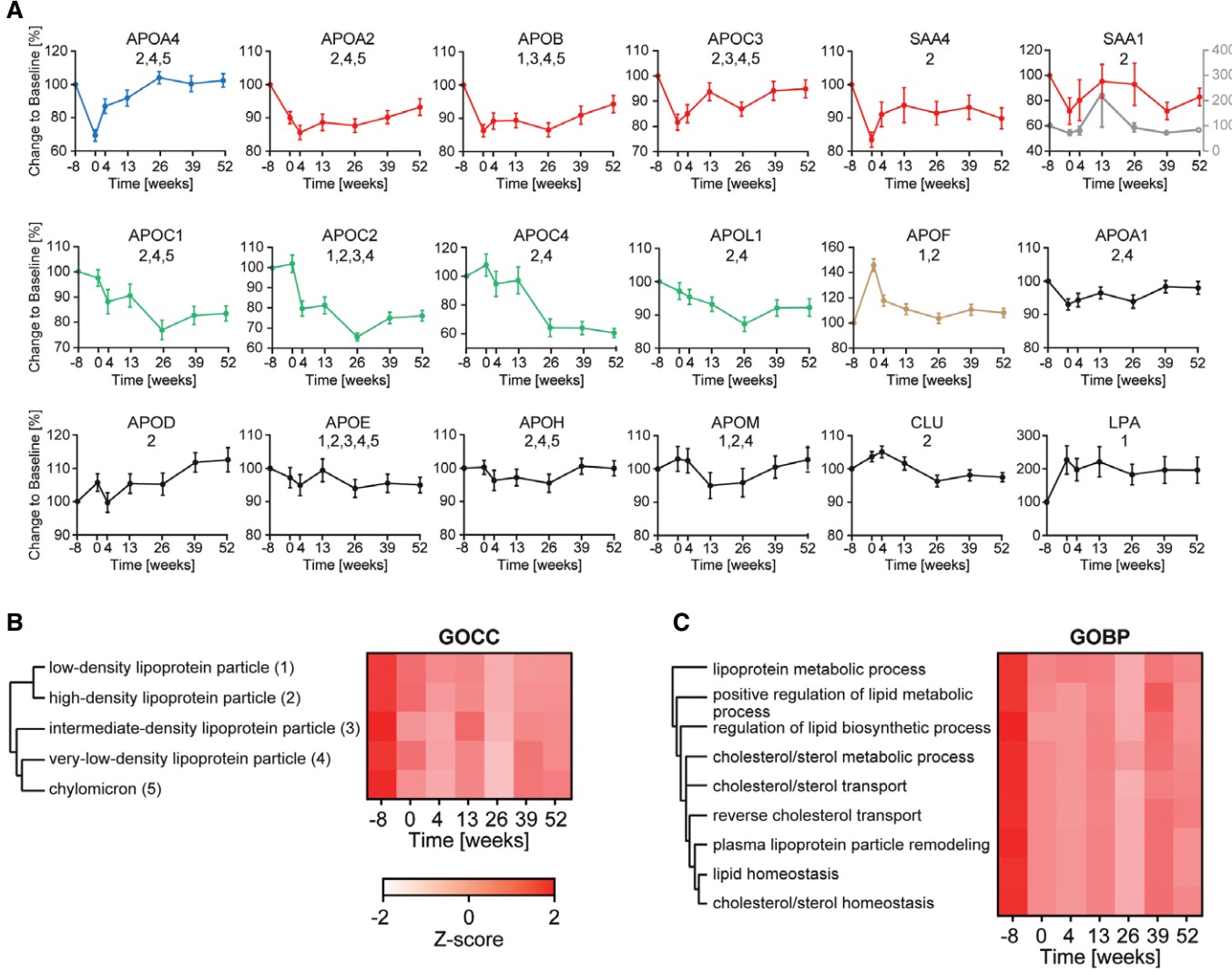

**Figure 7.   Effect of weight loss on the apolipoprotein family.**

A   The initial LFQ intensity before weight loss (week −8) was set to 100% to normalize protein abundance within each participant to account for individual-specific protein levels. Colors derive from clusters in Fig 4. The means are plotted with SEM as error bars over time. For the acute phase protein SAA1, the peak at week 13 was caused by very high levels in one individual (gray curve and right *y*-axis in the 6th panel). Excluding this individual results in the red curve. Numbers below the protein name refer to their presence in the lipoprotein particle in panel (C).

B   Annotation for gene ontology cellular component (GOCC) was *Z*-scored and filtered for main lipoprotein particles.

C   Gene ontology biological process (GOBP) annotations were filtered for the keywords lipid, lipoprotein, fat, and cholesterol, leading to the displayed group of GOBPs, which decreased due to weight loss.

before, but only with selected proteins and generally in smaller longitudinal studies (Crawford & Elisens, 2006; Kamstrup *et al*, 2008; Carlsson *et al*, 2010; Anderson, 2014). Our results suggest that it would be beneficial to determine baseline levels and variations in proteins by MS-based proteomics in even larger populations and to determine the underlying causes. Furthermore, these baseline levels could be determined in each patient in the context of precision medicine. This would enable patient-adjusted diagnostic tests and cutoff levels that take an individual's protein expression values as a reference rather than population-based reference intervals, which are used in clinical diagnostic tests today (Anderson, 2010). In the context of plasma proteome profiling, this

could be crucial for a proper interpretation of the plasma proteome in health and disease. Our results suggest that longitudinal plasma proteome profiles can circumvent problems associated with the natural variability of protein levels within and between individuals to a large degree.

Our study establishes that weight loss has a wide effect on the plasma proteome profile with a large proportion of quantified proteins changing significantly (93 proteins). Overall, we observed a strong difference between before and after weight loss followed by an adaption of the protein levels during the yearlong weight maintenance period. Many of the changes in the plasma proteome profile are readily explainable from the underlying biology and physiology.

    

For instance, levels of SERPINF1, which is secreted by adipocytes (Famulla *et al*, 2011), decrease with very high statistical significance, mirroring the loss of fat mass. SERPINF1 has already been associated with obesity before (Wang *et al*, 2008) but our study quantitatively establishes its fast downregulation in response to weight loss. This behavior and its consistency across the study population (in contrast to the known weight loss marker SHBG) could make SERPINF1 of possible interest in a clinical context.

Weight maintenance is a key challenge of any weight loss intervention. We therefore compared the initial plasma proteomes of poor (19 individuals), intermediate (13 individuals), and good (nine individuals) low-weight maintainers using body weight data from 2-year follow-up. This generated some interesting trends in plasma protein expression; however, these were not statistically significant (Table EV7). Nevertheless, future plasma proteomic studies may consider including such perspectives, potentially allowing identification of markers of individuals with a high probability of weight regain.

Monitoring the adaption of protein levels after weight loss yields new insights into the regulation of the plasma proteome. We identified several groups of highly significantly changed and functionally connected proteins with the same longitudinal behavior. Other proteins also clustered closely, but had no known functional connection. The tight cluster of four acute phase proteins, including CRP and SAA1, appears to represent systemic low-grade inflammation status in response to weight loss. The connection between high levels of CRP and obesity is well known (Yudkin *et al*, 1999; Selvin *et al*, 2007). Additionally, both proteins are associated with increased risk for cardiovascular diseases (CVD), where an increase of 10% of CRP levels leads to a 5.5% increase in CVD risk and a twofold increase of SAA1 to a 17% increase (Ridker *et al*, 2002; McEneny *et al*, 2015). In our study, weight loss induced a lowering of the individual's median levels of CRP by 35% and SAA1 by 44%, commonly accepted markers for CVD risk. The APOB/APOA1 ratio, another CVD risk marker, likewise decreased due to weight loss. In this way, plasma proteome profiling links previously established risk markers to weight loss. Apart from the specific aims of this study, plasma proteome profiling now provides the clinician with a new toolbox to investigate potentially important risk markers of CVD or other metabolic-related disease.

For the first time, MS-based plasma proteomics delivered a comprehensive picture of the response of proteins involved in lipid transport, including 18 apolipoproteins and other proteins that play a role in lipid metabolism. We found the expected correlations between LDL, HDL, total cholesterol, and triglycerides with constituent apolipoproteins, and these correlations may be even higher if the classical laboratory values would be reported more precisely. It would be interesting to investigate whether combinations of some of the top correlating proteins could be useful and robust risk markers. NRP1 and PRG4, which we identified by their longitudinal profiles and correlation with clinical parameters, are examples of promising candidates for further investigation. NRP1 was significantly increased at all time points after weight loss and the fact that NRP1 binds VEGF (Pellet-Many *et al*, 2008) makes it interesting to investigate a possible mechanism involving this interaction during weight loss. PRG4 was downregulated in response to weight loss. This proteoglycan lubricates articulating joints, and its presence in plasma may indicate tissue leakage. However, the strikingly strong correlation with LDL, triglyceride, and cholesterol levels may implicate PRG4 in lipid metabolism and in any case make it a potential biomarker related to LDL levels.

The anti-inflammatory proteins IL1RAP and CD14 increased after weight loss. This raises the possibility that the IL1 binding activity of IL1RAP results in antagonism of IL1 action and could play a role in lowering systemic inflammation (Smith *et al*, 2003). Levels of soluble IL1RAP and CD14 are known to be lower in obese individuals compared to controls (Bozaoglu *et al*, 2014; Laugerette *et al*, 2014). Our finding that IL1RAP and CD14 levels increased after weight loss therefore provides a possible mechanism that contributes to reduced low-grade inflammation.

We defined a panel of ten inflammation proteins and analyzed their correlation with BMI, to evaluate which individuals would benefit the most from weight loss based on a longitudinal inflammation profile. In our dataset, 39 of 42 individuals showed a clear positive effect in response to weight loss, and the three "non-profiting" individuals had increased inflammation levels apparently in part because of regain of weight. This indicates that the vast majority of obese individuals would profit from weight loss by improving their inflammatory profiles including known CVD and metabolic risk factors.

For further risk stratification of the cohort, we combined the inflammation panel with an insulin resistance panel, which was also defined by plasma proteome profiling. There was a high but not complete overlap of individuals in the two panels, pinpointing individuals with a high metabolic burden. Clearly, both these "high-risk" individuals and the other study participants greatly benefitted from weight loss and these effects persisted or even continued to improve over the 1-year weight maintenance period. Interestingly, the initial average values of the high-risk group in the inflammation and IR panel decreased to those of the other participants over the observational period.

From a clinical perspective, one could speculate that MS-based plasma proteomic may be used for patient stratification of obese subjects in a low and high metabolic burden profile, thereby providing a new diagnostic tool to intensify and optimize both pharmacological and non-pharmacological treatment of obese subjects with an elevated risk of cardiovascular disease. On the other hand, a global plasma proteomic analysis, as the one reported here, may target potential unknown metabolic regulators, thereby fostering future experiment setups by using a knockout approach of proteins of interest in rodents or in cell lines.

Incorporating an additional step of peptide fractionation would allow quantification of more than 1,000 proteins (Geyer *et al*, 2016), and adding multiplexing would further increase throughput and perhaps measurement precision. We envision such a capability to be available soon, which would enable routine measurements of studies such as this one in even greater depth and still in a reasonable amount of time. In the future, it would be interesting to use a workflow with even deeper coverage and higher throughput on a wide variety of clinical studies related to weight loss and other life style or pharmacological interventions. This would help to define novel risk markers and to disentangle correlations between them and existing clinical parameters. The resulting knowledge extracted from the plasma proteome could predict the individual gains expected from different interventions on the health or disease state.

# Materials and Methods

### Study design

Details of the weight loss study design are published elsewhere (Iepsen *et al*, 2015). In total, 58 obese study participants were recruited with the following inclusion criteria: healthy individuals with a BMI between 30 and 40 kg/m$^2$ and an age between 18 and 65 years. Excluded were participants with any acute or chronic illness other than obesity, any medical treatment with known effects on glucose and lipid metabolism, appetite or food intake, pregnancy or breast feeding and fasting plasma glucose levels of ≥7 mmol/l.

Study participants followed a weekly supervised very low-calorie powder diet (800 kcal per day; Cambridge Weight Plan, Corby, UK) for 8 weeks to achieve a weight loss of at least 7.5% of the initial body weight after 8 weeks (Riecke *et al*, 2010).

During the weight maintenance phase, the calorie intake of the study participants was restricted to the estimated daily energy needs subtracted by 600 kcal. In the case of weight gain, up to two meals a day during the weight maintenance period were allowed to be replaced by Cambridge Weight Plan products to ensure weight maintenance. Half of the participants also received 1.2 mg of liraglutide (daily) after weight loss. Both groups equally successfully maintained the weight loss with no significant change in weight from after weight loss to 1 year of weight maintenance (Iepsen *et al*, 2015). The difference in plasma proteomes between liraglutide-treated and non-treated individuals was subtle at our depth and accuracy of proteome measurement and was not further pursued in our analysis.

Blood samples were taken before weight loss (week −8), directly after (week 0) and at five time points during weight loss (weeks 4, 13, 26, 39, 52). Weight was measured at each visit, and further data for each participant were acquired for time points −8, 0, and 52.

The study was approved by the ethical committee in Copenhagen (reference number: H-4-210-134) and was performed in accordance with the Helsinki Declaration II and with ICH-GCP practice. Participation in the investigation was voluntary and the individuals could at any time retract their consent to participate. CliniclTrials.gov identifier: NCT02094183.

### Highly abundant protein depletion for building a matching library

We built up a matching library and used a depletion of the top 20 most abundant plasma proteins by a combination of two immuno-depletion kits (Nagaraj *et al*, 2012; Geyer *et al*, 2016). Plasma samples of three women and three men were obtained from a highly reliable reference blood bank (Plasma$^{Ref}$ Panels) from the Blutspendedienst des Bayerischen Roten Kreuzes. The Agilent Multiple Affinity Removal Spin Cartridge was used for the depletion of the top six highly abundant proteins (albumin, IgG, IgA, antitrypsin, transferrin, and haptoglobin), followed by ProteoPrep20 Plasma Immunodepletion Kit for the 20 highest abundant proteins (albumin, IgG, IgA, IgM, IgD, transferrin, fibrinogen, α2-macroglobulin, α1-antitrypsin, haptoglobin, α1-acid glycoprotein, ceruloplasmin, apolipoprotein A-I, apolipoprotein A-II, apolipoprotein B,

complement C1q, complement C3, complement C4, plasminogen, and prealbumin). Samples were depleted, digested, and measured in triplicate in the same way as the non-depleted sample set of the weight loss study.

### Sample preparation: protein digestion and in-StageTip purification

Sample preparation was carried out as described in Geyer *et al* (2016) and Kulak *et al* (2014) with the automated setup on an Agilent Bravo liquid handling platform. Plasma samples were diluted 1:10 with $_{dd}H_2O$ and 10 µl of the sample was mixed with 10 µl twofold concentrated SDC buffer. Reduction and alkylation were carried out at 95°C for 10 min. Trypsin and LysC (1:100 µg of enzyme to micrograms of protein ratio) were added to the mixture after a 5-min cooling step at room temperature. Digestion was performed at 37°C for 1 h. The digest was acidified by adding 40 µl of 1% trifluoroacetic acid (TFA) in isopropanol. An amount of 20 µg of peptides was loaded on two 14-gauge StageTip plugs, followed by the addition of 100 µl 1% trifluoroacetic acid (TFA) in isopropanol and strong mixing. The StageTips were centrifuged using a 3D-printed in-house-made StageTip centrifugal device at 1,500 *g*. After washing the StageTips two times using 100 µl 1% trifluoroacetic acid (TFA) in isopropanol and one time using 100 µl 0.2% TFA in $_{dd}H_2O$, purified peptides were eluted by 60 µl of elution buffer into autosampler vials. The collected material was completely dried using a SpeedVac centrifuge at 60°C (Eppendorf, Concentrator plus). Peptides were suspended in buffer A* (Kulak *et al*, 2014) and afterward sonicated (Branson Ultrasonics, Ultrasonic Cleaner Model 2510). It was not possible to obtain any peptides from one of the samples (# 44_1).

### Ultra-high-pressure liquid chromatography and mass spectrometry

Samples were measured using LC-MS instrumentation consisting of an EASY-nLC 1000 ultra-high-pressure system (Thermo Fisher Scientific), which was combined with a Q Exactive HF Orbitrap (Thermo Fisher Scientific) and a nano-electrospray ion source (Thermo Fisher Scientific) (Scheltema *et al*, 2014). Purified peptides were separated on 40-cm HPLC columns [ID: 75 µm; in-house packed into the tip with ReproSil-Pur C18-AQ 1.9 µm resin (Dr. Maisch GmbH)]. For each LC-MS/MS analysis, around 1 µg peptides was used for 45-min runs and for each fraction of the deep plasma dataset.

Peptides were loaded in buffer A (0.1% (v/v) formic acid) and eluted with a linear 18-min gradient of 5–20% of buffer B (0.1% (v/v) formic acid, 60% (v/v) acetonitrile), followed stepwise by a 12-min increase to 35% of buffer B, a 6 min to 50% of buffer B, 5.5-min increase to 98% of buffer B, followed by a 3.5-min wash of 98% buffer B at a flow rate of 350 nl/min. Column temperature was kept at 60°C by a Peltier element containing in-house-developed oven, and parameters were monitored in real time by the SprayQC software (Scheltema & Mann, 2012). MS data were acquired with a Top15 data-dependent MS/MS scan method (topN method). Target values for the full-scan MS spectra were $3 \times 10^6$ charges in the 300–1,650 m/z range with a maximum injection time of 55 ms and a resolution of 60,000 at m/z 200. Fragmentation of precursor ions was performed by higher-energy C-trap dissociation (HCD) with a

    

normalized collision energy of 27 eV. MS/MS scans were performed at a resolution of 30,000 at m/z 200 with an ion target value of $1 \times 10^5$ and a maximum injection time of 120 ms. Dynamic exclusion was set to 30 s to avoid repeated sequencing of identical peptides.

### Data analysis

Mass spectrometry raw files were analyzed by MaxQuant software version 1.5.3.23 (Cox & Mann, 2008), and peptide lists were searched against the human Uniprot FASTA database (version June 2015). A contaminants database by the Andromeda search engine (Cox *et al*, 2011) with cysteine carbamidomethylation as a fixed modification and N-terminal acetylation and methionine oxidations as variable modifications was used. We set the false discovery rate (FDR) to 0.01 for protein and peptide levels with a minimum length of seven amino acids for peptides, and the FDR was determined by searching a reverse database. Enzyme specificity was set as C-terminal to arginine and lysine as expected using trypsin and LysC as proteases, and a maximum of two missed cleavages were allowed. Peptide identification was performed with an initial precursor mass deviation up to 7 ppm and a fragment mass deviation of 20 ppm. The "match between run algorithm" in the MaxQuant quantification (Nagaraj *et al*, 2012) was performed after constructing a matching library consistent of depleted and all the undepleted plasma samples from the weight loss study. All proteins and peptides matching to the reversed database were filtered out. Label-free protein quantitation (LFQ) was performed with a minimum ratio count of 1 (Cox *et al*, 2014).

### Bioinformatics analysis

All bioinformatics analyses were done with the Perseus software of the MaxQuant computational platform (Cox & Mann, 2008; Tyanova *et al*, 2016). For statistical analysis of significantly changed proteins before (week −8) and directly after weight loss (week 0), a one-sample *t*-test was used with a false discovery rate of < 0.05 after Benjamini–Hochberg correction. We only considered highly significant proteins with a *P*-value of $P < 0.0005$ for the hierarchical clustering in Fig 4. For all correlation analyses, a false discovery rate of < 0.05 after Benjamini–Hochberg correction was applied.

All data needed for correlation analysis of classical clinical parameters like BMI, weight, levels of cholesterol, leptin, HDL, LDL, and triglycerides as well as HOMA-IR to MS-based proteomic acquired LFQ intensities are available for all study participants and time points (Table EV8).

### Data and materials availability

The MS-based proteomics data have been deposited to the ProteomeXchange Consortium via the PRIDE partner repository and are available via ProteomeXchange with the identifier PXD004242.

**Expanded View** for this article is available online.

### Acknowledgements

We thank all members of the Proteomics and Signal Transduction Group for help and discussions and in particular Korbinian Mayr, Igor Paron, and Gaby Sowa for technical assistance and Jürgen Cox for bioinformatic tools. The work carried out in this project was partially supported by the Max Planck Society for the Advancement of Science and by the Novo Nordisk Foundation (Grant NNF15CC0001).

### Author contributions

PEG designed, performed, and interpreted the MS-based proteomic analysis of patient plasma and wrote the paper and generated the figures. NJWA designed and interpreted the MS-based proteomic analysis of patient plasma and generated article text. ST provided statistical assistance, interpretation of the proteomic analysis, and revised the manuscript. NG performed MS-based proteomic analysis of patient plasma and revised the manuscript. EWI, JL, SM, JJH, and SST provided patient material and clinical data and revised the manuscript. MM designed and interpreted the MS-based proteomic analysis of patient plasma, supervised and guided the project, and wrote the paper.

### Conflict of interest

The authors declare that they have no conflict of interest.

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
