## [Review Process File · Molecular Systems Biology]

Proteomics Reveals the Effects of Sustained Weight Loss on the Human Plasma Proteome

Philipp E. Geyer, Nicolai J. Wewer Albrechtsen, Stefka Tyanova, Niklas Grassl, Eva W. Iepsen, Julie Lundgren, Sten Madsbad, Jens J. Holst, Signe S. Torekov, and Matthias Mann

Corresponding author: Matthias Mann, Max-Planck Institute of Biochemistry

Review timeline:	Submission date:	29 September 2016
	Editorial Decision:	03 November 2016
	Revision received:	17 November 2016
	Editorial Decision:	06 December 2016
	Revision received:	07 December 2016
	Accepted:	07 December 2016

Editor: Maria Polychronidu

Transaction Report:

1st Editorial Decision

03 November 2016

Thank you again for submitting your work to Molecular Systems Biology. We have now heard back from the three referees who agreed to evaluate your study. As you will see below, the reviewers raise a number of issues, which we would ask you to address in a revision. Please let me know in case you would like to further discuss any specific point.

If you feel you can satisfactorily deal with these points and those listed by the referees, you may wish to submit a revised version of your manuscript. Please attach a covering letter giving details of the way in which you have handled each of the points raised by the referees. A revised manuscript will be once again subject to review and you probably understand that we can give you no guarantee at this stage that the eventual outcome will be favorable.

REFeree REPORTS

Reviewer #1:

This is a useful report that documents the changes in a wide range of circulating human plasma proteins in response to initial weight loss and the maintenance of that weight loss

As befits the reputation of the senior author the technical aspects of the proteomic analysis are "state of the art" and I will not comment further on them

Given that this is, as the authors describe it, "by far the largest plasma proteomics study in a clinical context" to date it is somewhat disappointing that more thought was not given to designing a study that could have delivered some more interesting biological/biomedical insights. For example, in any

weight loss studies there will be those who regain and those to do not regain weight. It would have been fascinating to know if there were any elements of the proteomic profile of people, either before or after weight loss, that predicted those who would suffer from weight regain. As it stands, with its current design although this is a very nice and useful study which will provoke others in the field to do more, is largely documentary/descriptive in nature

The results section starts with a description of proteins whose concentrations vary widely between individuals but are stable within individuals. These include proteins for which the biological explanation of this behaviour is known and those for which it is unknown. However there is no new insight gained into the reasons for the behaviour in the case of the "unknown" ones

Weight loss decreased 63 and increased 30 protein levels. Some of these had not previously been examined in relation to weight change so the information is new
During weight maintenance some proteins that changed reverted towards their pre-weight loss levels and others remained at the altered level

The authors focus predominantly on apolipoproteins and inflammatory markers and confirm some known changes and add some new documentation

The authors write about the association of inflammatory markers with adiposity as if it were clear that the inflammatory mediators were playing a causative role in the adverse consequences of obesity. In fact, this is not at all clear. The best example is CRP, which is strongly associated with both BMI and with CV risk. However multiple large well-powered mendelian randomisation studies have unequivocally excluded a direct role for CRP in mediating adverse cardiovascular outcomes. i.e. it does reflect increased inflammation associated with increased adiposity but is merely a marker and has nothing to do with adverse cardiovascular outcomes. Indeed, to date, when it comes to insulin resistance or Type 2 diabetes there is no single inflammatory mediator for which there is any human genetic or pharmacological evidence supporting a causative role in disease pathogenesis

Much is made of correlations of markers with insulin resistance as assessed by HOMA IR. Unfortunately HOMA IR is itself only moderately well correlated with the gold standard measures of insulin resistance and, because it pertains to the fasting state, only really examines insulin sensitivity of the liver, not the skeletal muscle, the major site of insulin mediated glucose disposal. So what is being measured is the correlation of surrogate markers with something that is, itself, a surrogate. The leptin/adiponectin ratio has been previously reported to correlate better with clamp measures of insulin resistance than HOMA. If the authors have data on leptin (as they do for adiponectin) then it would be interesting to run that analysis.

In any case the authors need to be clearer that they are describing correlations not causative pathways and be careful not to overinterpret the meaning of the correlations

Reviewer #2:

In this study by Geyer and colleagues, the authors perform plasma discovery proteomics on 43 patients undergoing a clinical trial for weight loss across eight weeks, followed by longitudinal follow-up during a year of weight maintenance. Obesity is a major burden on our worldwide health services economy, making this study particularly relevant to a general audience.

The study is well written, the figures clearly convey the primary data and the data appear to be collected carefully and analyzed correctly. There are a few things to consider that might hopefully strengthen the study.

1. Inclusion of the protein LC-MS data (protein ID, patients, timepoint and relative quantification results) should be made available as part of the Supporting Information for this study, even though the primary data is deposited into PRIDE.
2. There are many transcriptomic, GWAS and NGS profiling studies on obesity, albeit from primarily adipose tissues, circulating blood cells and muscle. Greater comparisons to those studies is warranted.

3. There are many mouse models of obesity that could be used to both compare/contrast with the observations in the current study, or to test specific hypotheses regarding some of the protein markers observed herein. Such experiments may extend the study considerably, as it is currently quite observational in nature.

Reviewer #3:

Summary

The paper entitled "Proteomics Reveals the Effects of Sustained Weight Loss on the Human Plasma Proteome" by Geyer et al outlines a large-scale mass spectrometry-based plasma proteome profiling study using shotgun mass spectrometry to characterize plasma proteome changes during weight loss.

Based on a previously published study by Iepsen et al 2015, the plasma proteome from 52 obese individuals were monitored before and after a 12% reduction in body mass. In addition, the plasma proteome from 43 out of the 52 individuals were monitored 5 times per individual during a one-year period to assess the effects on the plasma proteome during maintained reduced weight. These 319 individual plasma samples were analyzed in quadruplicates using a previously published shotgun plasma profiling strategy quantifying 737 unique plasma proteins with a quantitative accuracy reflected by a mean Pearson correlation coefficient of 0.97 for the quadruplicate measurements.

The authors found that 93 proteins were significantly changed during weight loss although with small fold changes. The long-term effect of sustained weight loss is reflected in altered levels of inflammatory proteins.

General remarks

The paper represents an extensive quantitative proteome profiling study providing considerable insights into the changes occurring in the plasma proteome during weight loss. Overall it can be noted that the study reveal relatively minor changes in the plasma proteome, which were quantifiable due to the low CV's obtained with the used method as previously shown in the published paper by Geyer et al 2016 in Cell reports. The nature of the advance is the demonstration that plasma profiling using mass spectrometry is now feasible for more than a thousand samples, which can support future studies in the field. The study is of interest for proteomics researcher and for experts in the field of metabolic diseases.

Major points

Although the study is impressive in size and technical quality, the statistical analysis is not described in sufficient detail. For example it remains unclear if the authors adjusted for multiple hypothesis testing and how the technical replicates were used in the statistical analysis. Furthermore, the grouping of the proteins, based on the hierarchical clustering seems to be arbitrary at times. For example in the "inflammation panel" in Figure 6a, the dendrogram clearly shows subdivision into two groups. However, the line and the color-coding indicate that grouping incorrectly includes two patients from the adjacent group. The same phenomena can be observed in the Anti-IR and Pro-IR clustering. Based on this Figure and the text it is not clear how the authors subdivided the patients into a high Insulin Resistance group and a High Systemic Inflammation group. The authors need to revise the grouping of the patients and in addition include an improved description of how the patients were subdivided into these two groups. In a similar fashion, the protein clusters 3 and 4 shown in Figure 4 seem to be incorrect. Lastly, the selection of proteins in some Figures is based on p-value. It is questionable if the level of the p-value should be used as a factor for selecting proteins, which are all below the statistically significant cut-off value, as is shown in for example Figure 3a. The authors should clarify these points and add more information to the result section how the groups were selected. In addition, the authors should add a more elaborate section regarding the statistical analysis in the materials and method section.

The authors relied on samples from a study regarding weight loss published elsewhere (Iepsen et al., 2015). The study by Iepsen et al is a randomized treatment study, where it was shown that treatment with the GLP-1 receptor agonist (GLP-1RA) result in a significant decrease in weight loss compared to the controls during the weight maintenance period and a significant decrease in plasma

glucose levels and an increase in triglycerides levels. In the current form of the manuscript the authors only have a minor comment regarding the proteome comparison between treated and control patients in the materials and method section. The proteome change comparison between the treated and the control group should be shown more clearly in the paper both in the result section, preferably as a Figure, and in the discussion section. Although the results are subtle, this information is important when assessing the method and the other results presented in the manuscript.

As the authors only found subtle differences in the proteome comparison between the treated and non-treated sample groups the subdivision of patients based on treatment was omitted, leaving the study absent of a control group. Without the availability of a control group it is difficult to assess the consequences of the proteome changes during maintained weight loss. Would for example a control group with no weight loss have a distinct proteome profile compared to the study group? And how does the plasma proteome profiles compare to the plasma proteomes from lean individuals? It is advisable that the authors include a lean control group and a control group where blood plasma samples were taken from obese individual without weight loss and compare their proteome profiles with the study group. In this way, the magnitude of the observed changes could be better assessed.

The authors identify a group of patients that have elevated levels of inflammatory proteins and proteins that correlate to insulin resistance and refer to this group as "high risk". It is not entirely clear how the authors define these groups based on the presented data (see point above) and what exactly the authors mean by high-risk. Based on the data presented in Figure 6b the inflammatory panel is elevated in the patients with high BMI, which is expected as the proteins were originally selected based on their correlation with BMI (Fig 5a). These results imply that high BMI is the cause of the high levels of the inflammation panel and that the inflammation panel does not indicate high risk but rather high BMI. To test this the authors should investigate if there are any significantly regulated proteins between the high and low risk patients for the different time points. Also the authors should test the inflammation panel in the patients treated with GLP-1RA at the time points where there was a statistically significant difference in weight loss observed in the treated group, for example at week 13. Lastly, the authors state that 39 of the 42 patients "greatly benefitted from weight loss". There is not sufficient supporting data to conclude that the remaining 3 patients did not benefit from their weight loss. This statement should be rephrased or removed.

Minor points

The authors state that the total number of analyzed plasma proteomes is 1294. However, quadruplicate analysis of 319 samples equals 1276. In addition, on page four the author states both 319 and 318 samples.

1st Revision - authors' response

17 November 2016

Reviewer 1

This is a useful report that documentst the changes in a wide range of circulating human plasma proteins in response to initial weight loss and the maintenmance of that weight loss. As befits the reputation of the senior author the technical aspects of the proteomic analysis are "state of the art" and I will not comment further on them.

Answer: We thank the reviewer for the kind words regarding our proteomics technology and for the thoughtful and in-depth review of our manuscript. In the revised manuscript and below we have addressed the points of criticism that were raised, which we feel have improved this work.

Given that this is, as the authors descibe it, "by far the largest plasma proteomics study in a clinical context" to date it is somewhat disappointing that more thought was not given to designing a study that could have delivered some more intesting biological/biomedical insights. For example , in any weight loss studies there will be those who regain and those to do not regain weight. It would have been fascinating to know if ther were any elements of the proteomic profile of people, either before or after weight loss, that predicted those who would suffer fromweight regain. As it stands,with its current design although this is a very nice and useful study which will provoke others in the field to do more, is largely documentary/descriptive in nature

Regarding the design of the study we wished to focus on a metabolic perturbation of general interest, which also would have a good chance to lead to changes in the blood proteome that could be observed with our current technology. We feel this was ideally the case with the presented weight loss study, where we have very well characterized study participants and a relatively long observation period. Although we do not generate completely novel insights into the physiology of weight loss in this first study, we do think that we answer very fundamental questions about its impact on the plasma proteome, in particular discrimination between acute and long-term effects of weight loss, which is of great interest to many in the medical and proteomics communities.

That said, we completely agree with the reviewer that looking at the weight regain question is very appealing. At the time of submission we did not have the latest follow-up data (2-year data on body weight), but this has become available now. Therefore, as suggested by reviewer 1, we attempted to correlate the extent of weight regain in our study participants to their plasma proteomes. We grouped the individuals regarding their weight response into three groups: poor (19 individuals), intermediate (13 individuals) and good (9 individuals) low weight maintainer over the two years. We used 70% valid values and imputed based on a Gaussian distribution before we applied a support vector machine for classification feature optimization on the initial plasma proteomes before weight loss. This generated some trends, however, none of them are statically significant. This is not surprising given the relatively low numbers of participants involved. Furthermore, it is not clear that the information reflected in the plasma proteome before weight loss is sufficient to predict success in weight maintenance. In the discussion part of the revised manuscript, we briefly mention this topic.

The results section starts with a description of proteins whose concentration vary widely between individuals but are stable within individuals. These include proteins for which the biological explanation of this behaviour is known and those for which it is unknown. However there is no new insight gained into the reasons for the behaviour in the case of the "unknown" ones

We do agree with the reviewer that it would be valuable to know the molecular mechanism underlying the reported individual-specific protein levels. However, our intention was not to dissect each of the causation leading to these differences between study participants, but rather to demonstrate on a large scale -including several hundred proteins -that so many plasma proteins have an individual-specific level. As such these data may be useful for future studies that use any of these individual specific protein levels in a clinical context for the diagnosis of disease. Our results point in the direction of individual-specific diagnostic cut-off values, which could result in better disease diagnosis than it is possible by current population-based, 'one size fits all' cut-off values. We briefly mention this point in the revised manuscript.

In case of the "unknown" proteins, the reasons for their abundance levels and regulation will be as varied as for known proteins, ranging from gender, to genetics and -as we show in this manuscript to body weight. We believe that our resource will aid future research to disentangle the reasons behind individual-specific protein levels.

Weight loss decreased 63 and increased 30 protein levels. Some of these had not previously been examined in relation to weight change so the information is new. During weight maintenance some proteins that changed reverted towards their pre-weight loss levels and others remained at the altered level. The authors focus predominantly on apolipoproteins and inflammatory markers and confirm some known changes and add some new documentation.

The authors write about the association of inflammatory markers with adiposity as if it were clear that the inflammatory mediators were playing a causative role in the adverse consequences of obesity. In fact, this is not at all clear. The best example is CRP, which is strongly associated with both BMI and with CV risk. However multiple large well-powered mendelian randomisation studies have unequivocally excluded a direct role for CRP in mediating adverse cardiovascular outcomes. i.e it does reflect increased inflammation associated with increased adiposity but is merely a marker and has nothing to do with adverse cardiovascular outcomes. Indeed, to date, when it comes to insulin resistance or Type 2 diabetes there is no single inflammatory mediator for which there is any human genetic or pharmacological evidence supporting a causative role in disease pathogenesis

We thank the reviewer for pointing out that our interpretation may be implying a causative role for

the observed markers. Our manuscript does not focus on a single marker but merely investigates the global changes on the plasma proteome before, during and after a major physiological change (weight loss). As we do not have long-term follow up for these subjects, we now refrain from concluding that the inflammation panel we describe has any predictive value on type 2 diabetes or cardiovascular disease. In particular, to address the reviewer's concern, we have changed the paragraph in question to not any imply a causative connection between CVD outcome and CRP:

"In our study, weight loss induced a lowering of the individual's median levels of CRP by 35% and SAA1 by 44%, commonly accepted markers for CVD risk. The APOB/APOA1-ratio, another CVD risk marker, likewise decreased due to weight loss. In this way, plasma proteomic profiling links previously established risk markers to weight loss. Apart from the specific aims of this study, plasma proteome profiling now provides the clinician with a new toolbox to investigate potentially important risk markers of CVD or other metabolic related disease."

Much is made of correlations of markers with insulin resistance as assessed by HOMA IR. Unfortunately HOMA IR is itself only moderately well correlated with the gold standard measures of insulin resistance and, because it pertains to the fasting state, only really examines insulin sensitivity of the liver, not the skeletal muscle, the major site of insulin mediated glucose disposal. So what is being measured is the correlation of surrogate markers with something that is, itself, a surrogate. The leptin/adiponectin ratio has been previously reported to correlate better with clamp measures of insulin resistance than HOMA. If the authors have data on leptin (as they do for adiponectin) then it would be interesting to run that analysis.

We agree that HOMA IR is a surrogate marker of insulin resistance, but since it is used very broadly in the community, it is still interesting to analyze the correlation to the plasma proteome.

We do indeed have leptin levels for our study participants determined by ELISA assays and we now correlate this to the plasma proteome in the revised manuscript. The results enhance our finding for HOMA-IR and the connections of insulin resistance to low grade inflammation. Among the 80 proteins significantly correlating with leptin, many inflammation factors like CRP, SAA1, SAA4, C3 and CFH and APCS had a highly positive correlation (Table EV4). Remarkably, six of the proteins correlating with leptin are part of the ten-protein inflammation panel, which confirms the connection of insulin resistance and inflammation for another insulin resistance parameter. Regarding insulin resistance panels, the data were similar to what we reported using HOMA-IR: Leptin was positively correlated to all four proteins from the pro-HOMA-IR panel and two of the five proteins of the anti-HOMA-IR panel (NRP1 and APOF) were anti-correlated. These data enhance the connection of insulin resistance and inflammation and we mention them in the revised manuscript.

As the data for leptin are immuno assay-based concentration values, whereas for adiponectin we have MS-derived LFQ data, it is not straightforward to interpret their ratio. When we did this analysis, the results very strongly reflected the correlation to leptin alone.

In any case the authors need to be clearer that they are describing correlations not causative pathways and be careful not to overinterpret the meaning of the correlations

We agree and in the revised manuscript we more clearly discriminate between correlation and causality as described above for CVD risk.

Reviewer 2

In this study by Geyer and colleagues, the authors perform plasma discovery proteomics on 43 patients undergoing a clinical trial for weight loss across eight weeks, followed by longitudinal follow-up during a year of weight maintenance. Obesity is a major burden on our worldwide health services economy, making this study particularly relevant to a general audience.

The study is well written, the figures clearly convey the primary data and the data appear to be collected carefully and analyzed correctly. There are a few things to consider that might hopefully strengthen the study.

We thank the reviewer very much for these positive comments. In the revised manuscript we have incorporated some of the broader perspectives suggested.

Inclusion of the protein LC-MS data (protein ID, patients, timepoint and relative quantification results) should be made available as part of the Supporting Information for this study, even though the primary data is deposited into PRIDE.

As requested by the reviewer these data are now included as an extended view supplement (Table EV8), in which we supply all data needed for correlation analysis of classical clinical parameters like BMI, weight, levels of cholesterol, leptin, HDL, LDL and triglycerides as well as HOMA-IR to MS-based proteomic acquired LFQ intensities for all study participants and time points (Table EV8). The table is referenced from in the Material and Methods section in “Bioinformatic analysis”.

There are many transcriptomic, GWAS and NGS profiling studies on obesity, albeit from primarily adipose tissues, circulating blood cells and muscle. Greater comparisons to those studies is warranted.

Several omics approaches have been applied to dissect the underlying mechanisms of weight loss in humans and in the revised manuscript we have included a paragraph placing our study in a broader context. GWAS and NGS are not directly applicable to weight loss and maintenance studies as they reveal genetic predisposition and expression profiles in related tissues, respectively. Metabolomics, in contrast, is directly done on plasma and has been performed in the context of weight loss. We now provide this perspective and cite relevant papers as follows:

“Omics technologies have already been brought to bear on the study of obesity. GWAS studies have linked specific loci with genetic propensity for this trait whereas next transcriptome studies have investigated tissues such as fat, muscle or white blood cells. These studies are by their nature not directly connected to changes in protein levels in the plasma. In contrast, metabolomics has been performed on plasma in the context of weight loss, which demonstrated that several metabolic markers change after weight loss and similar to what reported here that individual specific levels seems to predominate (Newgard, 2016; Piccolo et al., 2015; Wahl et al., 2015). At the protein level, individual regulators of lipid transport have been studied in depth but our study provides a first proteomic view of changes in the plasma. Although not subject of this study, it would be interesting in the future to connect the different omics datasets to obtain a more comprehensive understanding of physiological changes during weight loss and maintenance.”

There are many mouse models of obesity that could be used to both compare/contrast with the observations in the current study, or to test specific hypotheses regarding some of the protein markers observed herein. Such experiments may extend the study considerably, as it is currently quite observational in nature.

We agree with the reviewer that including such an in-vivo experiments may further highlight the importance of potential new regulators of metabolism and the coupling between inflammation and metabolism. That said, this would require extensive work (minimum 1 year in our estimation) including generation of new genetic modified mice models and may not necessarily add information that is pertinent to the focus of our paper. Instead, we plan a follow up study (with a long multi-year observation period), which should in principle answer the same questions, however, in the more relevant human system.

Reviewer 3

The paper represents an extensive quantitative proteome profiling study providing considerable insights into the changes occurring in the plasma proteome during weight loss. Overall it can be noted that the study reveal relatively minor changes in the plasma proteome, which were quantifiable due to the low CV's obtained with the used method as previously shown in the published paper by Geyer et al 2016 in Cell reports. The nature of the advance is the demonstration that plasma profiling using mass spectrometry is now feasible for more than a thousand samples, which can support future studies in the field. The study is of interest for proteomics researcher and

for experts in the field of metabolic diseases.

Although the study is impressive in size and technical quality, the statistical analysis is not described in sufficient detail. For example it remains unclear if the authors adjusted for multiple hypothesis testing and how the technical replicates were used in the statistical analysis. Furthermore, the grouping of the proteins, based on the hierarchical clustering seems to be arbitrary at times. For example in the "inflammation panel" in Figure 6a, the dendrogram clearly shows subdivision into two groups. However, the line and the color-coding indicate that grouping incorrectly includes two patients from the adjacent group. The same phenomena can be observed in the Anti-IR and Pro-IR clustering. Based on this Figure and the text it is not clear how the authors subdivided the patients into a high Insulin Resistance group and a High Systemic Inflammation group. The authors need to revise the grouping of the patients and in addition include an improved description of how the patients were subdivided into these two groups. In a similar fashion, the protein clusters 3 and 4 shown in Figure 4 seem to be incorrect. Lastly, the selection of proteins in some Figures is based on p-value. It is questionable if the level of the p-value should be used as a factor for selecting proteins, which are all below the statistically significant cut-off value, as is shown in for example Figure 3a. The authors should clarify these points and add more information to the result section how the groups were selected. In addition, the authors should add a more elaborate section regarding the statistical analysis in the materials and method section.

We thank the reviewer for providing this constructive in-depth review of our manuscript. We provide more detail of our analyses, as requested, and have answered all the other points to the best of our ability below and in the revised manuscript.

Upon re-reading, we agree with the reviewer that the hierarchical clustering and the inclusion of study participants should be more clearly described. In the revised manuscript, we have clarified how the participants were assigned to the selected groups. Study participants 9 and 35 clustered in the lower cluster of the inflammation-panel, but in the initial manuscript they were assigned to the higher inflammation group because of their relatively high inflammation levels. We now leave these individuals in the lower inflammation cluster for a more objective analysis, which slightly changes the figure but does not change any of the conclusions.

The cohort is divided into four groups: Two with high and two with low levels of pro HOMA-IR correlating proteins. We have addressed the reviewer's concerns in the revised manuscript by a clarified description in the figure legend and a more detailed description of the statistical methods. Regarding the p-values, we only used proteins that showed a statistically significant correlation for the HOMA-IR and the inflammation panel. Moreover, figure 3a shows only the acute effect of weight loss, not the longer term follow up. Those differences are addressed in the following paragraph and in figure 4. We hope to have clarified this in the revised manuscript by a better description in the "Bioinformatics analysis" paragraph in "Materials and Methods".

The authors relied on samples from a study regarding weight loss published elsewhere (Iepsen et al., 2015). The study by Iepsen et al is a randomized treatment study, where it is shown that treatment with the GLP-1 receptor agonist (GLP-1RA) result in a significant decrease in weight loss compared to the controls during the weight maintenance period and a significant decrease in plasma glucose levels and an increase in triglycerides levels. In the current form of the manuscript the authors only have a minor comment regarding the proteome comparison between treated and control patients in the materials and method section. The proteome change comparison between the treated and the control group should be shown more clearly in the paper both in the result section, preferably as a Figure, and in the discussion section. Although the results are subtle, this information is important when assessing the method and the other results presented in the manuscript.

We did do the analysis of GLP-1 treated participants vs. non-treated. As the reviewer points out, the difference in weight loss, while significant, was relatively small (1.5 kg), which may partially be responsible for the fact that we did not see statistically significant changes between the two groups at the level of the plasma proteome with our current group size and technology. GLP-1 may instead impact the abundance of low-abundant peptides such as glucagon (stimulator of hepatic glucose production and is known to contribute to the diabetic hyperglycemia) which our study was not designed to capture. We now mention this in the results section as follows:

The participants of the study had been randomized to treatment with the incretin GLP-1, which showed a significant but small difference in weight loss (Iepsen et al., 2015). However, this difference did not correlate with statistically significant changes in our plasma proteome measurements.

As the authors only found subtle differences in the proteome comparison between the treated and non-treated sample groups the subdivision of patients based on treatment was omitted, leaving the study absent of a control group. Without the availability of a control group it is difficult to assess the consequences of the proteome changes during maintained weight loss. Would for example a control group with no weight loss have a distinct proteome profile compared to the study group? And how does the plasma proteome profiles compare to the plasma proteomes from lean individuals? It is advisable that the authors include a lean control group and a control group where blood plasma samples were taken from obese individual without weight loss and compare their proteome profiles with the study group. In this way, the magnitude of the observed changes could be better assessed.

Each study participant was followed longitudinal over more than one year and in this way they each constitute their own reference. The advantage of this kind of control is that it allows disentangling the observations from individual-specific protein levels and natural variation within the study cohort – i.e. we compensate partly for genetic and other differences between participants. For these reasons, we purposefully did not select a cross-sectional design. This is not to say that such a comparison would not be interesting: on the contrary, we agree with the reviewer that comparing cohorts of obese and lean subjects would be a worthwhile endeavor for the future. For the present manuscript, this would both take the focus away from the weight loss and would also be impractical due to the time lines involved in finding perfectly matched study participants that would agree to be monitored for a year without intervention.

The authors identify a group of patients that have elevated levels of inflammatory proteins and proteins that correlate to insulin resistance and refer to this group as "high risk". It is not entirely clear how the authors define these groups based on the presented data (see point above) and what exactly the authors mean by high-risk. Based on the data presented in Figure 6b the inflammatory panel is elevated in the patients with high BMI, which is expected as the proteins were originally selected based on their correlation with BMI (Fig 5a). These results imply that high BMI is the cause of the high levels of the inflammation panel and that the inflammation panel does not indicate high risk but rather high BMI. To test this the authors should investigate if there are any significantly regulated proteins between the high and low risk patients for the different time points. Also the authors should test the inflammation panel in the patients treated with GLP-1RA at the time points where there was a statistically significant difference in weight loss observed in the treated group, for example at week 13. Lastly, the authors state that 39 of the 42 patients "greatly benefitted from weight loss". There is not sufficient supporting data to conclude that the remaining 3 patients did not benefit from their weight loss. This statement should be rephrased or removed.

These concerns have been addressed in the revised manuscript in conjunction with those of reviewer 1, who correctly pointed out that we cannot infer causality from association on the basis of risk markers (see above). In particular, we removed statements that imply causality. Moreover, we have rephrased the statement regarding the remaining 3 patients as requested by the reviewer. Finally, we have more clearly described and discussed the data shown in Figure 6B in relation to BMI (Figure 5A).

The authors state that the total number of analyzed plasma proteomes is 1294. However, quadruplicate analysis of 319 samples equals 1276. In addition, on page four the author states both 319 and 318 samples.

We thank the reviewer for catching this seeming inconsistency, which are addressed in the revised manuscript. In this study we measured 319 samples in quadruplicates and a further 18 depleted samples as a matching library, which results in a total of 1294 plasma proteomes. We now added a better description in the Results section in addition to the one in the legend of figure 1.

This is a useful report that documents the changes in a wide range of circulating human plasma proteins in response to initial weight loss and the maintenance of that weight loss. As befits the reputation of the senior author the technical aspects of the proteomic analysis are "state of the art" and I will not comment further on them.

Answer: We thank the reviewer for the kind words regarding our proteomics technology and for the thoughtful and in-depth review of our manuscript. In the revised manuscript and below we have addressed the points of criticism that were raised, which we feel have improved this work.

Given that this is, as the authors describe it, "by far the largest plasma proteomics study in a clinical context" to date it is somewhat disappointing that more thought was not given to designing a study that could have delivered some more interesting biological/biomedical insights. For example, in any weight loss studies there will be those who regain and those who do not regain weight. It would have been fascinating to know if there were any elements of the proteomic profile of people, either before or after weight loss, that predicted those who would suffer from weight regain. As it stands, with its current design although this is a very nice and useful study which will provoke others in the field to do more, is largely documentary/descriptive in nature.

Regarding the design of the study we wished to focus on a metabolic perturbation of general interest, which also would have a good chance to lead to changes in the blood proteome that could be observed with our current technology. We feel this was ideally the case with the presented weight loss study, where we have very well characterized study participants and a relatively long observation period. Although we do not generate completely novel insights into the physiology of weight loss in this first study, we do think that we answer very fundamental questions about its impact on the plasma proteome, in particular discrimination between acute and long-term effects of weight loss, which is of great interest to many in the medical and proteomics communities.

That said, we completely agree with the reviewer that looking at the weight regain question is very appealing. At the time of submission we did not have the latest follow-up data (2-year data on body weight), but this has become available now. Therefore, as suggested by reviewer 1, we attempted to correlate the extent of weight regain in our study participants to their plasma proteomes. We grouped the individuals regarding their weight response into three groups: poor (19 individuals), intermediate (13 individuals) and good (9 individuals) low weight maintainers over the two years. We used 70% valid values and imputed based on a Gaussian distribution before we applied a support vector machine for classification feature optimization on the initial plasma proteomes before weight loss. This generated some trends, however, none of them are statically significant. This is not surprising given the relatively low numbers of participants involved. Furthermore, it is not clear that the information reflected in the plasma proteome before weight loss is sufficient to predict success in weight maintenance. In the discussion part of the revised manuscript, we briefly mention this topic.

The results section starts with a description of proteins whose concentrations vary widely between individuals but are stable within individuals. These include proteins for which the biological explanation of this behaviour is known and those for which it is unknown. However there is no new insight gained into the reasons for the behaviour in the case of the "unknown" ones.

We do agree with the reviewer that it would be valuable to know the molecular mechanism underlying the reported individual-specific protein levels. However, our intention was not to dissect each of the causations leading to these differences between study participants, but rather to demonstrate on a large scale -including several hundred proteins -that so many plasma proteins have an individual-specific level. As such these data may be useful for future studies that use any of these individual specific protein levels in a clinical context for the diagnosis of disease. Our results point in the direction of individual-specific diagnostic cut-off values, which could result in better disease diagnosis than it is possible by current population-based, 'one size fits all' cut-off values. We briefly mention this point in the revised manuscript.

In case of the "unknown" proteins, the reasons for their abundance levels and regulation will be as varied as for known proteins, ranging from gender, to genetics and -as we show in this manuscript to body weight. We believe that our resource will aid future research to disentangle the reasons behind individual-specific protein levels.

Weight loss decreased 63 and increased 30 protein levels. Some of these had not previously been examined in relation to weight change so the information is new. During weight maintenance some proteins that changed reverted towards their pre-weight loss levels and others remained at the altered level. The authors focus predominantly on apolipoproteins and inflammatory markers and confirm some known changes and add some new documentation.

The authors write about the association of inflammatory markers with adiposity as if it were clear that the inflammatory mediators were playing a causative role in the adverse consequences of obesity. In fact, this is not at all clear. The best example is CRP, which is strongly associated with both BMI and with CV risk. However, multiple large well-powered Mendelian randomisation studies have unequivocally excluded a direct role for CRP in mediating adverse cardiovascular outcomes, i.e. it does reflect increased inflammation associated with increased adiposity but is merely a marker and has nothing to do with adverse cardiovascular outcomes. Indeed, to date, when it comes to insulin resistance or Type 2 diabetes there is no single inflammatory mediator for which there is any human genetic or pharmacological evidence supporting a causative role in disease pathogenesis.

We thank the reviewer for pointing out that our interpretation may be implying a causative role for the observed markers. Our manuscript does not focus on a single marker but merely investigates the global changes on the plasma proteome before, during and after a major physiological change (weight loss). As we do not have long-term follow up for these subjects, we now refrain from concluding that the inflammation panel we describe has any predictive value on type 2 diabetes or cardiovascular disease. In particular, to address the reviewer's concern, we have changed the paragraph in question to not imply a causative connection between CVD outcome and CRP:

"In our study, weight loss induced a lowering of the individual's median levels of CRP by 35% and SAA1 by 44%, commonly accepted markers for CVD risk. The APOB/APOA1-ratio, another CVD risk marker, likewise decreased due to weight loss. In this way, plasma proteomic profiling links previously established risk markers to weight loss. Apart from the specific aims of this study, plasma proteome profiling now provides the clinician with a new toolbox to investigate potentially important risk markers of CVD or other metabolic related disease."

Much is made of correlations of markers with insulin resistance as assessed by HOMA-IR. Unfortunately HOMA-IR is itself only moderately well correlated with the gold standard measures of insulin resistance and, because it pertains to the fasting state, only really examines insulin sensitivity of the liver, not the skeletal muscle, the major site of insulin mediated glucose disposal. So what is being measured is the correlation of surrogate markers with something that is, itself, a surrogate. The leptin/adiponectin ratio has been previously reported to correlate better with clamp measures of insulin resistance than HOMA-IR. If the authors have data on leptin (as they do for adiponectin) then it would be interesting to run that analysis.

We agree that HOMA-IR is a surrogate marker of insulin resistance, but since it is used very broadly in the community, it is still interesting to analyze the correlation to the plasma proteome.

We do indeed have leptin levels for our study participants determined by ELISA assays and we now correlate this to the plasma proteome in the revised manuscript. The results enhance our finding for HOMA-IR and the connections of insulin resistance to low grade inflammation. Among the 80 proteins significantly correlating with leptin, many inflammation factors like CRP, SAA1, SAA4, C3 and CFH and APCS had a highly positive correlation (Table EV4). Remarkably, six of the proteins correlating with leptin are part of the ten-protein inflammation panel, which confirms the connection of insulin resistance and inflammation for another insulin resistance parameter. Regarding insulin resistance panels, the data were similar to what we reported using HOMA-IR: Leptin was positively correlated to all four proteins from the pro-HOMA-IR panel and two of the five proteins of the anti-HOMA-IR panel (NRP1 and APOF) were anti-correlated. These data enhance the connection of insulin resistance and inflammation and we mention them in the revised manuscript.

As the data for leptin are immuno assay-based concentration values, whereas for adiponectin we have MS-derived LFQ data, it is not straightforward to interpret their ratio. When we did this analysis, the results very strongly reflected the correlation to leptin alone.

In any case the authors need to be clearer that they are describing correlations not causative pathways and be careful not to overinterpret the meaning of the correlations

We agree and in the revised manuscript we more clearly discriminate between correlation and causality as described above for CVD risk.

2nd Editorial Decision

06 December 2016

Thank you for submitting your revised manuscript. We have now heard back from reviewer #3 who was asked to evaluate the study. The reviewer thinks that the study is now suitable for publication and did not have any further comments on the work.

Before we formally accept the study for publication, I would like to ask you to address some remaining editorial issues.

3rd Editorial Decision

07 December 2016

Thank you for addressing these remaining minor editorial issues. I am pleased to inform you that your paper has been accepted for publication.

Corresponding Author Name: Matthias Mann

Manuscript Number: MSB-16-7357